# HiCI: Hierarchical Construction–Integration for Long-Context Attention

**Xiangyu Zeng**[1]  **Qi Xu**[2]  **Yunke Wang**[1]  **Chang Xu**[1]

 github.com/zengxyyu/HiCI

## Abstract

Long-context language modeling is commonly framed as a scalability challenge of token-level attention, yet local-to-global information structuring remains largely implicit in existing approaches. Drawing on cognitive theories of discourse comprehension, we propose **HiCI** (Hierarchical Construction–Integration), a hierarchical attention module that constructs segment-level representations, integrates them into a shared global context, and broadcasts both to condition segment-level attention. We validate HiCI through parameter-efficient adaptation of LLaMA-2 (7B and 13B) and Qwen3-8B with only ∼4–5% additional parameters, extending context to 100K/64K tokens for LLaMA-2-7B/13B and to 48K tokens for Qwen3-8B. Across language modeling, retrieval, and instruction-following benchmarks, HiCI yields consistent improvements over strong baselines, including matching proprietary models on topic retrieval and surpassing GPT-3.5-Turbo-16K on code comprehension. These results demonstrate the effectiveness of explicit hierarchical structuring as an inductive bias for long-context modeling.

## 1. Introduction

Large language models (LLMs) have achieved remarkable success across a wide range of natural language tasks, yet their ability to process long sequences remains fundamentally constrained by limited context windows (Vaswani et al., 2017; Brown et al., 2020; Li et al., 2025; Wang et al., 2025; 2026). Long-context modeling poses two fundamental challenges: (1) *efficiency*—the quadratic complexity of self-attention leads to prohibitive computational and memory costs as sequence length increases; and (2) *effectiveness*—the ability to accept longer inputs does not necessarily yield reliable modeling of long-range dependencies (Hsieh et al., 2024; Liu et al., 2024) . Reconciling these two requirements has emerged as a central challenge in long-context language modeling.

Recent work has progressed along two complementary lines. The first pursues **positional length generalization**: techniques such as PI, YaRN, and PoSE (Chen et al., 2023; Peng et al., 2024; Zhu et al., 2024) extend the usable context window by interpolating, rescaling, or simulating position indices, yet leave the attention operator—and its $\mathcal{O}(n^2)$ complexity—unchanged. The second focuses on **attention efficiency and architectural scalability**, comprising two broad families. Sparse and grouped attention (Beltagy et al., 2020; Zaheer et al., 2020; Chen et al., 2024) reduces cost by restricting token connectivity, with global interactions approximated via global tokens or layer-wise multi-hop mixing from shifted grouping. Recurrent and memory-augmented architectures (Dai et al., 2019; Bulatov et al., 2022; Munkhdalai et al., 2024; He et al., 2025) model cross-segment dependencies through compressed state propagation, but sequential processing limits parallelism and long-range information may be attenuated through the compression bottleneck. While effective for length generalization or efficiency, these approaches offer limited inductive bias for explicitly organizing long-context information into a local-to-global hierarchy that guides attention.

Cognitive theories of discourse comprehension offer a principled lens on this limitation. The Construction-Integration model (Kintsch, 1988; 1998) characterizes text understanding as a hierarchical process in which local representations are first constructed from input segments and subsequently integrated—via constraint satisfaction—into a coherent global representation. Complementarily, Global Workspace Theory (Baars, 1993; Dehaene & Naccache, 2001) posits that specialized processors operate in parallel, with information gaining access to a shared workspace being *broadcast* globally, achieving wide availability and exerting top-down influence on subsequent processing. This broadcast mechanism finds support in hierarchical cortical processing (Felleman & Van Essen, 1991), where top-down sig-

---

[1]School of Computer Science, University of Sydney, Sydney, Australia [2]Westlake University, Hangzhou, China. Correspondence to: Chang Xu <c.xu@sydney.edu.au>.

*Proceedings of the 43$^{rd}$ International Conference on Machine Learning*, Seoul, South Korea. PMLR 306, 2026. Copyright 2026 by the author(s).

nals modulate lower-level representations. Taken together, these perspectives motivate a hierarchical inductive bias: *local construction* of segment-level representations, *global integration* into a shared context, and *top-down broadcast* to condition subsequent attention.

Guided by this principle, we propose **HiCI** (**Hi**erarchical **C**onstruction–**I**ntegration), a hierarchical attention module that instantiates *construction, integration, and broadcast* within Transformer attention. HiCI structures attention computation through three stages. **Local construction** extracts segment-level representations via cross-attention with a shared set of learnable query slots. **Global integration** aggregates these local representations into a compact shared context through multi-view statistical pooling and attention-based weighting. **Top-down broadcast** prepends both global and local representations to each segment's key–value sequence, conditioning token-level attention on hierarchical context while preserving segment-parallel computation.

We apply HiCI to pretrained LLaMA-2 models (Touvron et al., 2023) and Qwen3-8B (Yang et al., 2025), combining position interpolation (Chen et al., 2023) for context extension with FlashAttention-2 (Dao, 2024) for efficient long-sequence computation. Following LongLoRA's parameter-efficient recipe (Chen et al., 2024), we freeze the backbone and train only LoRA adapters, embedding and normalization layers, together with the proposed HiCI module. Despite adding only ∼4–5% parameters during training, this enables context extension to 100K/64K tokens for LLaMA-2-7B/13B and to 48K tokens for Qwen3-8B. At inference, HiCI is optional: it can be applied during prefill to reduce time-to-first-token latency, or omitted in favour of standard full attention.

Extensive experiments on language modeling (PG-19 (Rae et al., 2020), Proof-pile (Azerbayev et al., 2022)), retrieval (passkey and topic), and instruction-following (Long-Bench (Bai et al., 2024b)) benchmarks demonstrate that HiCI consistently improves performance over strong baselines across a wide range of tasks and context lengths. HiCI achieves 100% passkey accuracy (Mohtashami & Jaggi, 2023) within the 32K training regime and maintains substantially higher accuracy under direct extrapolation, attains the best topic-retrieval (Li et al., 2023) accuracy among the evaluated open-source models, and achieves higher accuracy than GPT-3.5-Turbo-16K (Achiam et al., 2023) on the Code category of LongBench (+9.7%). Ablation studies further identify global integration as the dominant contributor within HiCI, and show that explicit hierarchical conditioning is essential beyond grouped attention alone.

In summary, our contributions are:

- We propose HiCI, a hierarchical attention module that instantiates construction–integration–broadcast as an explicit inductive bias for long-context modeling in Transformers.

- We show that HiCI generalises across model families and scales, requiring only ∼4–5% additional parameters to extend LLaMA-2-7B/13B to 100K/64K tokens and Qwen3-8B to 48K, while consistently improving perplexity and downstream performance on retrieval and instruction-following tasks.

- Ablation studies identify global integration as the dominant component, and reveal an optimal compact slot budget. Attention visualizations further show that deeper layers increasingly attend to global representations, indicating emergent hierarchical information routing.

## 2. Related Work

### 2.1. Efficient Attention Mechanisms

The quadratic complexity of self-attention has motivated extensive research on efficient alternatives. **Sparse attention** restricts the attention pattern to reduce computation: Longformer (Beltagy et al., 2020) employs sliding windows augmented with task-specific global tokens, BigBird (Zaheer et al., 2020) combines local, global, and random attention to achieve linear complexity with theoretical guarantees, and LongNet (Ding et al., 2023) uses dilated attention with exponentially increasing receptive fields across heads. **Linear attention** (Katharopoulos et al., 2020) approximates softmax via kernel decomposition, enabling $O(n)$ complexity. However, kernel-based approximations exhibit degraded performance on retrieval-intensive tasks (Arora et al., 2024), and predefined sparsity patterns limit adaptability to diverse long-range dependencies.

### 2.2. Context Window Extension for LLMs

LLMs are typically pre-trained with fixed context lengths (e.g., 4,096 for Llama-2), and context extension has been pursued through *positional scaling* and *efficient long-context adaptation*. **Positional encoding** methods modify RoPE-style position representations to improve length extrapolation. Position Interpolation (PI) (Chen et al., 2023) rescales position indices and relies on substantial continued training to adapt to longer contexts. Subsequent schemes such as YaRN (Peng et al., 2024) and LongRoPE (Ding et al., 2024) introduce frequency-aware or non-uniform scaling, reducing the amount of long-context continued training relative to PI. These methods address *where* to attend but retain quadratic complexity and do not alter how attention organizes context. **Training and adaptation** methods address long-context fine-tuning with varying efficiency. Early work such as Focused Transformer (Tworkowski et al., 2023) employs

specialized training objectives, but remains computationally intensive (128 TPUs). More efficient alternatives have since emerged: LongLoRA (Chen et al., 2024) combines shifted sparse attention with LoRA, enabling 100k context on 8×A100; PoSE (Zhu et al., 2024) simulates long positions within fixed windows; LongAlign (Bai et al., 2024a) accelerates training via packing strategies. Despite substantially reducing adaptation cost, these methods lack an explicit mechanism for organizing and globally sharing contextual information. HiCI builds upon LongLoRA while introducing hierarchical context organization, constructing local-to-global abstractions that condition token-level attention (Section 3).

### 2.3. Segment-based Long-context Modeling

The $\mathcal{O}(L^2)$ cost of self-attention motivates segment-wise processing, trading direct cross-segment interaction for efficiency. Existing approaches differ in how they restore this connectivity. **Recurrence-based methods** propagate information through sequential state updates across segments. Transformer-XL (Dai et al., 2019) caches hidden states from prior segments and attends to them as extended context, RMT (Bulatov et al., 2022) transmits learnable memory tokens across segment boundaries, and Block-Recurrent Transformer (Hutchins et al., 2022) combines block-level recurrence with attention for improved parallelism. Despite their effectiveness, sequential dependencies limit parallel training and risk information attenuation over long distances. **Compression-based methods** summarize past segments into fixed-capacity representations. Compressive Transformer (Rae et al., 2020) learns to compress older memories, while Infini-attention (Munkhdalai et al., 2024) incrementally updates a compressive state via linear attention. These approaches bound memory but sacrifice fine-grained fidelity (Xu et al., 2026a;b). **Hierarchical methods** construct multi-level abstractions. HMT (He et al., 2025) maintains a memory hierarchy with segment summarization, Block Transformer (Ho et al., 2024) separates global block-level and local token-level attention into distinct modules, bypassing token-level KV cache for faster inference, and EM-LLM (Fountas et al., 2025) segments via Bayesian surprise inspired by episodic memory. In addition to these explicit mechanisms, LongLoRA (Chen et al., 2024) partitions attention into local groups and enables implicit interaction via shifted grouping across heads. In summary, existing segment-based methods restore cross-segment connectivity at the cost of parallelism, fidelity, or explicit semantic organization. Motivated by Construction–Integration (Kintsch, 1988) and Global Workspace Theory (Baars, 1993), HiCI addresses these limitations: segment-local representations are constructed via cross-attention, integrated into global context, and both are concatenated with original tokens in KV space—enabling parallel, semantically explicit conditioning

over long contexts.

### 2.4. Global Workspace Architectures and Latent Representations

Global Workspace Theory (GWT) (Baars, 1993) posits that cognition arises from a shared broadcast mechanism that integrates and disseminates information across specialised modules. VanRullen & Kanai (2021) translate this framework into design principles for deep networks; Goyal et al. (2022); Zeng et al. (2024) instantiate a shared workspace for inter-module communication in modular networks, while Hong et al. (2024) extend the motif to concept-centric transformer representations. Complementary to architectural coordination, recent work explores whether discrete tokens are the appropriate medium for language model computation. Coconut (Hao et al., 2025) replaces chain-of-thought tokens with continuous latent states for multi-step reasoning, whereas Large Concept Models (Barrault et al., 2024) predict sentence-level embeddings instead of next tokens. HiCI is inspired by the GWT intuition of constructing and integrating global context, but differs in architectural scope: rather than coordinating distinct functional modules or concept abstractions, it performs cross-segment context aggregation within a single transformer layer for long-context language modeling. Unlike latent-space approaches, HiCI preserves standard next-token autoregressive prediction; its global context vectors are auxiliary attention prefixes injected as keys and values rather than latent reasoning states or prediction targets.

## 3. Hierarchical Construction–Integration Attention

We present HiCI, a lightweight attention module that instantiates a cognitively motivated inductive bias for long-context modeling. HiCI organizes attention computation into three stages—local construction, global integration, and top-down broadcast—mirroring the hierarchical process of human discourse comprehension.

### 3.1. Overview

Standard self-attention induces pairwise interactions among all $T$ tokens, resulting in $\mathcal{O}(T^2)$ computational complexity (Vaswani et al., 2017). A widely adopted alternative is segmented attention, which partitions the input into fixed-length segments and restricts attention to within-segment interactions, reducing the complexity to $\mathcal{O}(T \cdot S)$. However, such formulations lack an explicit mechanism for propagating information across segments. HiCI addresses this limitation through structured context conditioning: it dynamically constructs compact local and global representations from the input and injects them back into each block's attention

computation.

Given an input sequence $X \in \mathbb{R}^{T \times d}$, assuming $T$ is divisible by the segment length $S$, we partition it into $N = T/S$ segments $X_1, \ldots, X_N$ and proceed as follows (Figure 1):

1. **Local Construction** (§3.2): For each segment $X_i \in \mathbb{R}^{S \times d}$, cross-attention with $M$ learnable query slots extracts a local representation $L_i \in \mathbb{R}^{M \times d}$.

2. **Global Integration** (§3.3): The local representations $\{L_i\}_{i=1}^N$ are aggregated into a shared global context $G \in \mathbb{R}^{K \times d}$ via multi-view statistical pooling followed by attention-based weighting.

3. **Top-down Broadcast** (§3.4): The global context $G$ and segment-specific abstraction $L_i$ are prepended to the key–value sequence of each segment $X_i$, conditioning token-level updates on hierarchical context while preserving parallelism across segments.

Throughout, the cardinalities $M$ and $K$ are fixed constants independent of the sequence length $T$.

## 3.2. Local Construction

The first stage performs *local construction*, distilling each input segment $X_i \in \mathbb{R}^{S \times d}$ into a compact representation $L_i \in \mathbb{R}^{M \times d}$, where $M \ll S$ is a small, sequence-length-independent constant, consistent with the limited capacity of human working memory (Miller, 1956; Cowan, 2001).

**Bottleneck Cross-Attention.** We introduce $M$ learnable slot vectors $L_{\text{slot}} \in \mathbb{R}^{M \times d}$, shared across all segments, which serve as queries attending to segment tokens via multi-head cross-attention. To improve parameter efficiency and induce abstraction, attention is computed in a low-dimensional subspace $\mathbb{R}^{d_b}$ with $d_b \ll d$.

Formally, for each segment $X_i \in \mathbb{R}^{S \times d}$, the local representation $L_i \in \mathbb{R}^{M \times d}$ is computed as

$$\tilde{L}_i = \text{softmax}\left( \frac{(L_{\text{slot}} W_Q^\ell)(X_i W_K^\ell)^\top}{\sqrt{d_k}} \right) (X_i W_V^\ell), \quad (1)$$

$$L_i = \tilde{L}_i W_O^\ell, \quad (2)$$

where $\{W_Q^\ell, W_K^\ell, W_V^\ell\} \in \mathbb{R}^{d \times d_b}$ and $W_O^\ell \in \mathbb{R}^{d_b \times d}$ are learned projections, with $H$ attention heads of dimension $d_k = d_b/H$.

The bottleneck $(M, d_b)$ defines a fixed-capacity interface that favors salient segment-level structure over fine-grained token detail. Aggregating the resulting $\{L_i\}_{i=1}^N$ yields $L \in \mathbb{R}^{N \times M \times d}$ for subsequent integration. A formal treatment of this constraint is given in Appendix A.

## 3.3. Global Integration

Given the stacked local representations $L \in \mathbb{R}^{N \times M \times d}$, the global integration stage consolidates segment-level information into a compact global context $G \in \mathbb{R}^{K \times d}$, where a small $K$ reflects the capacity constraints of a global workspace (Baars, 1993).

**Multi-View Statistical Aggregation.** We collapse the segment and slot dimensions of $L \in \mathbb{R}^{N \times M \times d}$ into a single axis, yielding $\mathcal{L} \in \mathbb{R}^{(NM) \times d}$, and compute five complementary statistics over this axis:

$$\boldsymbol{\mu} = \frac{1}{NM} \sum_{j=1}^{NM} \mathcal{L}_j, \quad (3)$$

$$\boldsymbol{\mu}^+ = \max_j \mathcal{L}_j, \quad \boldsymbol{\mu}^- = \min_j \mathcal{L}_j, \quad (4)$$

$$\boldsymbol{\sigma} = \sqrt{\frac{1}{NM} \sum_{j=1}^{NM} (\mathcal{L}_j - \boldsymbol{\mu})^2}, \quad (5)$$

$$\hat{\boldsymbol{\mu}} = \boldsymbol{\mu}/\|\boldsymbol{\mu}\|_2. \quad (6)$$

Each statistic lies in $\mathbb{R}^d$ and captures a complementary aspect of the aggregated representations: $\boldsymbol{\mu}$ reflects central tendency, $\boldsymbol{\mu}^+$ and $\boldsymbol{\mu}^-$ capture element-wise extremal activations, $\boldsymbol{\sigma}$ measures dispersion, and $\hat{\boldsymbol{\mu}}$ encodes directional information independent of magnitude via $\ell_2$-normalization.

**Shared Compression.** We organize the five statistics into a matrix

$$\mathbf{Z} = \left[ \boldsymbol{\mu}; \, \boldsymbol{\mu}^+; \, \boldsymbol{\mu}^-; \, \boldsymbol{\sigma}; \, \hat{\boldsymbol{\mu}} \right] \in \mathbb{R}^{5 \times d}, \quad (7)$$

where each row corresponds to one statistical view. Rather than learning separate projections, we apply a shared two-stage compression $\phi : \mathbb{R}^d \to \mathbb{R}^{d_b}$:

$$\tilde{\mathbf{Z}} = \phi(\mathbf{Z}) = \psi_b \circ \psi_c(\mathbf{Z}), \quad (8)$$

where $\psi_c(\cdot) = \text{LayerNorm}(\cdot \, W_c)$ with $W_c \in \mathbb{R}^{d \times d_s}$, and $\psi_b(\cdot) = \text{LayerNorm}(\cdot \, W_b)$ with $W_b \in \mathbb{R}^{d_s \times d_b}$. The intermediate bottleneck $d_s < d_b \ll d$ induces abstraction via an information bottleneck (Tishby et al., 2000), while parameter sharing enforces consistent compression across heterogeneous statistical views.

**Attention-Based Selection.** We introduce $K$ learnable query vectors $Q_G \in \mathbb{R}^{K \times d_b}$ that attend to the compressed statistics via multi-head cross-attention. Formally,

$$G_c = \text{softmax}\left( \frac{(Q_G W_Q^g)(\tilde{\mathbf{Z}} W_K^g)^\top}{\sqrt{d_b/H}} \right) (\tilde{\mathbf{Z}} W_V^g), \quad (9)$$

where $\{W_Q^g, W_K^g, W_V^g\} \in \mathbb{R}^{d_b \times d_b}$ are learned projections with $H$ attention heads and $d_b$ as in §3.2. The output is

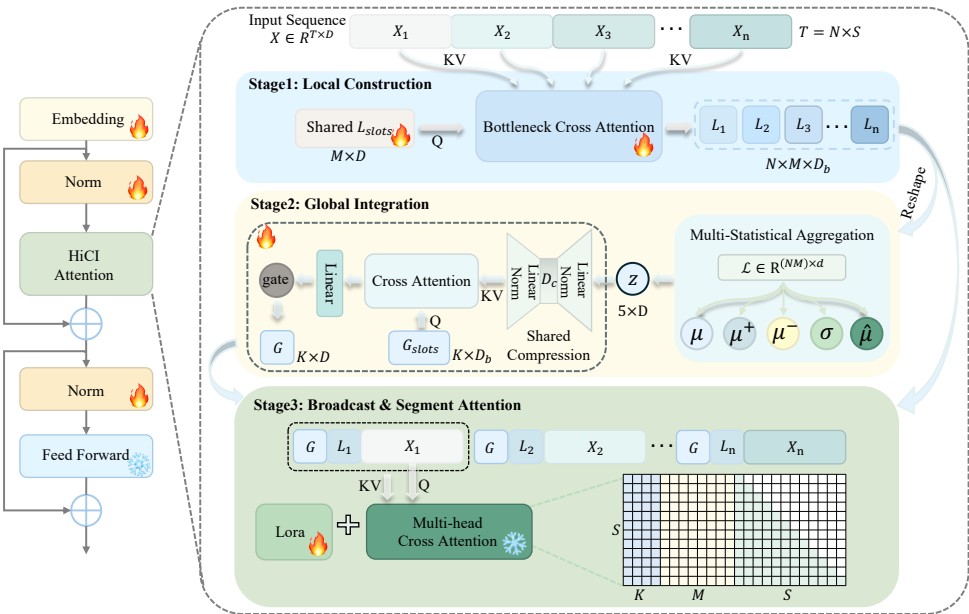

*Figure 1.* **Overview of HiCI.** *Left:* HiCI integrated into a Transformer block; trainable components are highlighted. *Right:* HiCI constructs hierarchical context through three stages. **(1) Local Construction:** the input sequence is partitioned into $N$ segments, and cross-attention with $M$ learnable query slots extracts a local representation $L_i$ from each segment. **(2) Global Integration:** local representations $\{L_i\}_{i=1}^N$ are aggregated into a shared global context $G$ via multi-view statistical pooling and attention-based weighting. **(3) Top-down Broadcast:** $G$ and $L_i$ are prepended to each segment's key–value sequence, conditioning attention on hierarchical context while preserving parallelism across segments. At inference, HiCI is optionally applied during prefill, while autoregressive decoding uses standard attention.

then projected back to the model dimension with a learnable gate:

$$G = G_c W_{\exp} \cdot \alpha, \quad \alpha = \ln(1 + e^\beta), \qquad (10)$$

where $W_{\exp} \in \mathbb{R}^{d_b \times d}$ and $\beta \in \mathbb{R}$ is a learnable scalar. The constraint $\alpha > 0$ ensures stable scaling of the global context. The resulting $G \in \mathbb{R}^{K \times d}$ serves as the global context for top-down broadcast (§3.4).

### 3.4. Top-down Broadcast

The final stage performs top-down broadcast, conditioning segment-level attention on both the globally integrated context $G$ and the corresponding local abstraction $L_i$.

For each segment $X_i \in \mathbb{R}^{S \times d}$, we form a context-augmented sequence by concatenating the global and local representations with the segment tokens:

$$[\, G;\, L_i;\, X_i \,] \in \mathbb{R}^{(K+M+S) \times d}.$$

The augmented sequence is projected into the key–value space as

$$\tilde{K}_i = [\, G;\, L_i;\, X_i \,] W_K^b, \qquad \tilde{V}_i = [\, G;\, L_i;\, X_i \,] W_V^b, \tag{11}$$

where $W_K^b, W_V^b \in \mathbb{R}^{d \times d}$.

Queries are derived exclusively from segment tokens as $Q_i = X_i W_Q^b$, where $W_Q^b \in \mathbb{R}^{d \times d}$.

Attention over the augmented context yields a context-conditioned update:

$$\tilde{X}_i = \mathrm{softmax}\left(\frac{Q_i \tilde{K}_i^\top}{\sqrt{d/H}}\right) \tilde{V}_i \in \mathbb{R}^{S \times d}, \tag{12}$$

where $H$ is the number of attention heads.

Since each segment attends to its augmented context independently, all $N$ segments can be processed in parallel. The refined segments are concatenated to form the output:

$$\tilde{X} = \mathrm{Concat}(\tilde{X}_1, \ldots, \tilde{X}_N) \in \mathbb{R}^{T \times d}. \tag{13}$$

By jointly attending over all $K+M+S$ positions under a unified softmax, each token integrates global, local, and segment-level context, implementing top-down modulation (see Appendix A for analysis).

## 4. Experiments

In this section, we evaluate the effectiveness of HiCI across language modeling(§4.2), retrieval (§4.3), and downstream benchmarks (§4.4), followed by ablation studies (§4.5). Additional attention analysis is given in the Appendix C.

### 4.1. Experimental Setup

**Models.** We evaluate HiCI on pretrained LLaMA-2 (Touvron et al., 2023) 7B/13B and Qwen3-8B (Yang et al., 2025)

*Table 1.* Perplexity (↓) on PG-19 and Proof-pile test sets for LLaMA-2-7B/13B and Qwen3-8B continually pre-trained on RedPajama for 1k training steps. LLaMA-2 models are trained with context lengths ranging from 8K to 100K and evaluated up to 100K; Qwen3-8B extends its original 32K context window to 48K and is evaluated up to 48K, with an additional 500-step variant reported.

| Base Model | Train | Method | PG-19 | | | | | | | Proof-pile | | | | | | |
|---|---|---|---|---|---|---|---|---|---|---|---|---|---|---|---|---|
| | | | 2K | 4K | 8K | 16K | 32K | 64K | 100K | 2K | 4K | 8K | 16K | 32K | 64K | 100K |
| LLaMA-2-7B | 4k | LLaMA-2-7B | 7.49 | 7.15 | $>10^2$ | $>10^2$ | $>10^2$ | $>10^2$ | $>10^2$ | 3.21 | 2.91 | $>10^2$ | $>10^2$ | $>10^2$ | $>10^2$ | $>10^2$ |
| | | ChunkLLaMA | 7.49 | 7.15 | 6.98 | 6.96 | 7.08 | 15.15 | – | 3.21 | 2.91 | 2.75 | 2.69 | 2.70 | 2.75 | – |
| | 8K | LongLoRA | 7.70 | 7.35 | 7.14 | – | – | – | – | 3.20 | 2.91 | 2.72 | – | – | – | – |
| | | HiCI | 7.27 | 7.01 | 6.93 | – | – | – | – | 3.07 | 2.82 | 2.65 | – | – | – | – |
| | 16K | LongLoRA | 7.65 | 7.28 | 7.02 | 6.86 | – | – | – | 3.17 | 2.87 | 2.66 | 2.51 | – | – | – |
| | | HiCI | 7.53 | 7.21 | 6.96 | 6.84 | – | – | – | 3.15 | 2.84 | 2.61 | 2.47 | – | – | – |
| | 32K | LongLoRA | 8.29 | 7.83 | 7.54 | 7.35 | 7.22 | – | – | 3.35 | 3.01 | 2.78 | 2.61 | 2.50 | – | – |
| | | HiCI | 7.87 | 7.50 | 7.26 | 7.09 | 7.11 | – | – | 3.21 | 2.87 | 2.71 | 2.58 | 2.49 | – | – |
| | 100K | LongLoRA | 8.38 | 7.90 | 7.57 | 7.33 | 7.16 | 7.06 | 7.04 | 3.36 | 3.01 | 2.78 | 2.60 | 2.58 | 2.57 | 2.52 |
| | | HiCI | 7.81 | 7.72 | 7.45 | 7.26 | 7.08 | 6.97 | 6.95 | 3.27 | 2.86 | 2.73 | 2.54 | 2.48 | 2.46 | 2.43 |
| LLaMA-2-13B | 4k | LLaMA-2-13B | 6.86 | 6.55 | $>10^2$ | $>10^2$ | $>10^2$ | $>10^2$ | $>10^2$ | 3.04 | 2.78 | 76.05 | $>10^2$ | $>10^2$ | $>10^2$ | $>10^2$ |
| | | ChunkLLaMA | 6.86 | 6.55 | 6.37 | 6.35 | 6.46 | 14.36 | – | 3.04 | 2.78 | 2.62 | 2.55 | 2.58 | 2.64 | – |
| | 8K | LongLoRA | 7.03 | 6.73 | 6.58 | – | – | – | – | 3.04 | 2.77 | 2.60 | – | – | – | – |
| | | HiCI | 6.68 | 6.46 | 6.34 | – | – | – | – | 2.91 | 2.69 | 2.52 | – | – | – | – |
| | 16K | LongLoRA | 7.05 | 6.70 | 6.47 | 6.31 | – | – | – | 3.03 | 2.74 | 2.55 | 2.41 | – | – | – |
| | | HiCI | 6.95 | 6.65 | 6.43 | 6.28 | – | – | – | 2.99 | 2.73 | 2.53 | 2.40 | – | – | – |
| | 32K | LongLoRA | 7.05 | 6.70 | 6.47 | 6.31 | 6.20 | – | – | 3.03 | 2.74 | 2.55 | 2.41 | 2.32 | – | – |
| | | HiCI | 6.94 | 6.56 | 6.39 | 6.25 | 6.17 | – | – | 2.94 | 2.68 | 2.40 | 2.35 | 2.26 | – | – |
| | 64K | LongLoRA | 7.63 | 7.21 | 6.94 | 6.75 | 6.62 | 6.53 | – | 3.05 | 2.76 | 2.57 | 2.42 | 2.32 | 2.25 | – |
| | | HiCI | 7.40 | 7.06 | 6.81 | 6.62 | 6.47 | 6.39 | – | 2.96 | 2.63 | 2.38 | 2.31 | 2.20 | 2.17 | – |
| | | | 2K | 4K | 8K | 16K | 32K | 48K | – | 2K | 4K | 8K | 16K | 32K | 48K | – |
| Qwen3-8B | 32k | Qwen3-8B | 13.26 | 12.58 | 12.09 | 11.72 | 12.76 | 11.32 | – | 3.24 | 2.94 | 2.73 | 2.59 | 2.49 | 2.46 | – |
| | 48K | HiCI (500s) | 11.71 | 11.06 | 10.59 | 10.24 | 9.98 | 9.89 | – | 3.01 | 2.73 | 2.54 | 2.41 | 2.32 | 2.29 | – |
| | | HiCI | 11.46 | 10.84 | 10.38 | 10.06 | 9.82 | 9.73 | – | 2.95 | 2.68 | 2.50 | 2.37 | 2.30 | 2.26 | – |

models, extending their context windows using Position Interpolation (Chen et al., 2023) to 100K/64K tokens for LLaMA-2-7B/13B and 48K for Qwen3-8B.

**Training.** Following LongLoRA (Chen et al., 2024), we perform two-stage LoRA fine-tuning: continued pretraining on RedPajama (Computer, 2023) with the next-token prediction objective, then instruction tuning on LongAlpaca-12k (Chen et al., 2024), training only the HiCI module, LoRA adapters, embeddings, and normalization layers. Optimization is performed with AdamW ($\beta_1$=0.9, $\beta_2$=0.95, weight decay 0), using a learning rate of $2\times10^{-5}$ for the backbone and $2\times10^{-4}$ for HiCI with a 20-step linear warmup. Unless otherwise specified, we train for 1,000 steps with per-device batch size 1 and gradient accumulation 8, yielding an effective batch size of 64. All experiments use bf16 precision with DeepSpeed ZeRO-2 (Rasley et al., 2020) and Flash-Attention2 (Dao, 2024), running on 8×H100 GPUs for LLaMA-2 and 8×H200 GPUs for Qwen3-8B; full hyperparameter details are provided in Appendix B.1.

**Evaluation.** We adopt the two-stage evaluation protocol of

LongLoRA. Stage 1 assesses long-context language modeling and retrieval: we report perplexity on PG-19 (Rae et al., 2020) and Proof-pile (Azerbayev et al., 2022) using a sliding window with stride 256 (Press et al., 2021) and the same hierarchical attention as training, along with passkey retrieval (Mohtashami & Jaggi, 2023) and topic retrieval (Li et al., 2023). Stage 2 evaluates downstream instruction-following on LongBench (Bai et al., 2024b) under two inference modes: standard full attention and HiCI attention during prefill.

### 4.2. Language Modeling

We evaluate perplexity on PG-19 (Rae et al., 2020) and Proof-pile (Azerbayev et al., 2022), comparing HiCI against LongLoRA (Chen et al., 2024) and ChunkLLaMA (An et al., 2024) for LLaMA-2-7B/13B (training lengths 8K–100K, evaluation up to 100K), and against the base model for Qwen3-8B under context extension from 32K to 48K, as reported in Table 1. The longest-context settings (100K for LLaMA-2-7B and 64K for LLaMA-2-13B) use DeepSpeed

Stage-3 (Rajbhandari et al., 2020) with adjusted group configurations; details are provided in Appendix B.1. HiCI reduces perplexity across all evaluation lengths for both model families, indicating that hierarchical context organization enhances language modeling quality beyond context extension alone. For LLaMA-2-7B/13B, HiCI consistently outperforms LongLoRA across model scales and training lengths, with ChunkLLaMA serving as a training-free reference. The performance gap is more pronounced at shorter evaluation lengths: for the 100K-trained LLaMA-2-7B, HiCI achieves a 6.8% reduction at 2K evaluation, compared to 1.3% at 100K, suggesting better preservation of short-range modeling quality. For Qwen3-8B, HiCI achieves a 14% perplexity reduction on PG-19 and 8% on Proof-pile at 48K, demonstrating that the hierarchical context mechanism transfers to models with stronger native context; where a 500-step variant already captures most of the gain.

### 4.3. Retrieval-based Evaluation

**Topic Retrieval.** We evaluate on the LongChat topic retrieval task (Li et al., 2023), which requires identifying a target topic from multi-turn dialogues spanning 3K–16K tokens. As shown in Table 2, while closed-source models such as GPT-4o-mini-128K (Hurst et al., 2024) and Claude-3.5-Sonnet-200K (Anthropic, 2024) achieve perfect accuracy, open-source alternatives show notable degradation: models with shorter context windows (e.g., ChatGLM2-6B-8k (Du et al., 2022), MPT-30B-Chat-8k (MosaicML, 2023), and Llama-3-8B-Instruct-8K (Grattafiori et al., 2024)) fail beyond their training length, and even MPT-7B-StoryWriter-65K (MosaicML, 2023) achieves only 0.28–0.46 across all lengths. In contrast, HiCI-13B-16K achieves the best accuracy among open-source models, matching 100% up to 13K and reaching 0.94 at 16K, compared to 0.90 for LongChat-13B-16K (Li et al., 2023) and 0.86 for LongLoRA-13B-16K (Chen et al., 2024). We conjecture that HiCI's stability is driven by a hierarchical inductive bias: segment-level construction learns content-dependent representations, while global integration forms position-invariant contextual representations, reducing sensitivity to where evidence appears in the sequence.

**Passkey Retrieval.** We evaluate passkey retrieval following Mohtashami & Jaggi (2023), where models are required to locate and output a random passkey embedded within long distractor text. For each context length, we conduct 10 trials with randomized passkey values and insertion positions. Figure 2 compares HiCI-7B-32K, LongLoRA-7B-32K (Chen et al., 2024), and the base LLaMA-2-7B model (Touvron et al., 2023). Within the 32K training regime, HiCI achieves 100% retrieval accuracy across all evaluated lengths, whereas LongLoRA exhibits non-monotonic behavior with accuracy fluctuating between 80% and 100%, and the base LLaMA-2-7B model (Touvron et al.,

*Table 2.* Topic retrieval accuracy on LongChat (Li et al., 2023). We compare HiCI against both proprietary models and open-source long-context LLMs across 3K–16K context lengths. HiCI-13B-16K matches proprietary model performance up to 13K and outperforms all open-source baselines at 16K.

| Model | 3K | 6K | 10K | 13K | 16K |
|---|---|---|---|---|---|
| GPT-4o-mini-128K | 1.00 | 1.00 | 1.00 | 1.00 | 1.00 |
| Claude-3.5-Sonnet-200K | 1.00 | 1.00 | 1.00 | 1.00 | 1.00 |
| MPT-30B-Chat-8K | 0.96 | 1.00 | 0.76 | – | – |
| ChatGLM2-6B-8K | 0.88 | 0.46 | 0.02 | 0.02 | 0.02 |
| MPT-7B-StoryWriter-65K | 0.46 | 0.46 | 0.28 | 0.34 | 0.36 |
| LongChat-13B-16K | 1.00 | 1.00 | 1.00 | 0.98 | 0.90 |
| LongLoRA-13B-16K[†] | 1.00 | 0.96 | 1.00 | 0.98 | 0.86 |
| Llama-3-8B-Instruct-8K | 1.00 | 1.00 | 0.00 | 0.00 | 0.00 |
| **HiCI-13B-16K (Ours)** | **1.00** | **1.00** | **1.00** | **1.00** | **0.94** |

[†] Evaluated with official LoRA weights.

2023), constrained by its native 4K context window, fails to retrieve passkeys beyond this length. To assess length extrapolation, we extend the maximum context at inference time to 56K using position interpolation (PI) (Chen et al., 2023), without any additional fine-tuning, following Chen et al. (2024). Beyond the 32K, both fine-tuned models exhibit degradation, consistent with the known sensitivity of RoPE-based positional encoding to out-of-distribution positions. Notably, HiCI degrades more gracefully, maintaining 40–60% retrieval accuracy over the 33K–56K range, compared to LongLoRA's 10–30% accuracy under the same setting. These results suggest that HiCI's training-time inductive bias may yield representations more robust to position extrapolation.

### 4.4. Downstream Tasks

**LongBench.** LongBench (Bai et al., 2024b) is a bilingual benchmark comprising 21 tasks across six categories, with average input lengths of 5K–15K tokens. We perform context extension on RedPajama (4K→16K) followed by instruction tuning on LongAlpaca-12k (Chen et al., 2024), using LoRA (Hu et al., 2022) with trainable embedding and normalization layers as in LongLoRA (Chen et al., 2024). We evaluate two inference modes: HiCI with standard full attention, and HiCI[†] which applies training-consistent hierarchical attention during prefill to reduce time-to-first-token latency. As shown in Table 3, HiCI outperforms LongLoRA across most categories, achieving 33.2% overall (+2.6%). The gains are particularly pronounced on Single-Document QA (**+7.4%**) and Chinese tasks (**+11.8%**), suggesting that the hierarchical inductive bias benefits both localized comprehension and cross-lingual transfer. HiCI[†], despite using hierarchical attention during prefill, maintains competitive performance (32.9%) and surpasses all baselines including the proprietary model GPT-3.5-Turbo-16K (Achiam et al., 2023) on both Summarization (24.6%, +0.7%) and Code

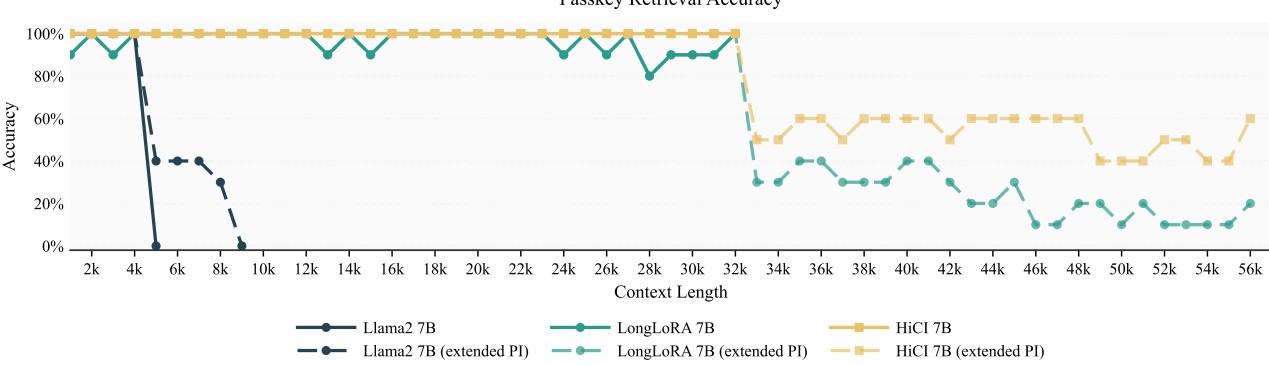

*Figure 2.* Passkey retrieval accuracy for LongLoRA-7B, HiCI-7B (both fine-tuned at 32K), and base LLaMA-2-7B. HiCI achieves 100% accuracy within the training length and extrapolates more gracefully to 56K via position interpolation without additional fine-tuning.

*Table 3.* Results (%) on LongBench (Bai et al., 2024b) benchmark. Best in **bold**, second underlined. ↑ indicates improvement over LongLoRA-7B-16K (direct baseline) and ↑ highlights top-2 gains for each HiCI variant.

| Model | Single-Doc QA | Multi-Doc QA | Summ | Few-shot | Synthetic | Code | Overall EN | ZH | All |
|---|---|---|---|---|---|---|---|---|---|
| GPT-3.5-Turbo-16k | **45.1** | **36.2** | 23.9 | **57.6** | **51.0** | 54.1 | **44.0** | **44.5** | **44.7** |
| Llama2-7B-chat-4k | 21.7 | 18.2 | 18.5 | 49.9 | 4.1 | 48.1 | 31.0 | 14.3 | 26.8 |
| LongChat-7B-32k | 28.8 | 20.3 | 22.5 | 50.8 | 13.0 | 54.1 | 34.3 | 23.9 | 31.6 |
| Vicuna-v1.5-7B-16k | 31.8 | 18.8 | 23.2 | 56.8 | 5.3 | 47.3 | 31.9 | 26.4 | 30.5 |
| LongLoRA-7B-16k | 23.7 | 25.0 | 20.9 | 54.2 | 12.0 | 55.8 | 36.8 | 10.9 | 30.6 |
| **HiCI-7B-16k** | 31.1$^{\uparrow 7.4}$ | 26.8$^{\uparrow 1.8}$ | 23.6$^{\uparrow 2.7}$ | 57.1$^{\uparrow 2.9}$ | 5.8 | 62.0$^{\uparrow 6.2}$ | 36.4 | 22.7$^{\uparrow 11.8}$ | 33.2$^{\uparrow 2.6}$ |
| **HiCI-7B-16k**$^{\dagger}$ | 29.9$^{\uparrow 6.2}$ | 24.5 | **24.6**$^{\uparrow 3.7}$ | 57.0$^{\uparrow 2.8}$ | 6.1 | **63.8**$^{\uparrow 8.0}$ | 35.8 | 23.4$^{\uparrow 12.5}$ | 32.9$^{\uparrow 2.3}$ |

$^{\dagger}$ Applies training-consistent HiCI attention during inference prefill.

(63.8%, **+9.7%**) tasks. This indicates that the learned hierarchical structure transfers robustly even under efficient inference.

### 4.5. Ablation Studies

We systematically evaluate HiCI along two axes: component contribution and representation cardinality. All experiments use LLaMA-2-7B as the base model and evaluate under standard full attention. Segment-size sensitivity is analyzed in Appendix B.3.

**Component and Cardinality Analysis.** To quantify the contribution of each HiCI component, we train variants under 8K context for 1,000 steps and evaluate on PG-19 and Proof-pile test sets. As shown in Table 4, removing global integration (w/o G) incurs nearly twice the degradation of removing local construction (w/o L), revealing that cross-segment aggregation contributes more substantially than within-segment compression. This asymmetry is corroborated by attention visualizations in Appendix C. The Only Group baseline—grouped attention without hierarchical modules—yields markedly inferior performance, underscoring that explicit integration is indispensable be-

*Table 4.* Component and cardinality ablation for HiCI fine-tuned on LLaMA-2-7B at 8K context for 1,000 steps.

| Variant | L | G | B | PG19 4K | 8K | Proof-pile 4K | 8K |
|---|---|---|---|---|---|---|---|
| HiCI | ✓ | ✓ | ✓ | **7.01** | **6.93** | **2.82** | **2.65** |
| w/o G | ✓ | ✗ | ✓ | 7.25 | 7.04 | 2.95 | 2.78 |
| w/o L | ✗ | ✓ | ✓ | 7.13 | 6.99 | 2.86 | 2.69 |
| Only Group | ✗ | ✗ | ✗ | 8.01 | 7.54 | 3.26 | 2.97 |
| $M=5,\ K=3$ | ✓ | ✓ | ✓ | 7.15 | 6.98 | 2.85 | 2.68 |
| $M=8,\ K=4$ | ✓ | ✓ | ✓ | **7.01** | **6.93** | **2.82** | **2.65** |
| $M=9,\ K=7$ | ✓ | ✓ | ✓ | 7.10 | 6.96 | 2.86 | 2.69 |

yond attention sparsification alone. For representation capacity, ($M{=}8, K{=}4$) attains optimal performance, aligning with Miller's $7 \pm 2$ working memory bound (Miller, 1956); smaller capacities $(5, 3)$ prove insufficient, while larger ones $(9, 7)$ compromise length generalization.

### 5. Limitations

HiCI currently improves efficiency primarily during training; autoregressive decoding still relies on standard full

attention with a growing KV cache, and extending the hierarchical mechanism to the decoding phase remains an open direction. A second limitation concerns models with very long native context windows (e.g., 128K tokens or beyond). HiCI is primarily motivated by settings where pre-training context is limited; whether it remains beneficial for models already pre-trained at such lengths has not been empirically studied. Finally, improving long-context efficiency may lower the barrier to large-scale handling of highly coherent long-form content, which could be misused in adversarial settings or involve increasingly sensitive user-provided data. These broader risks require system-level safeguards beyond the scope of this work.

## 6. Conclusion

We presented HiCI, a hierarchical attention framework that decomposes long-context attention into local construction, global integration, and top-down broadcast. With only 4–5% additional trainable parameters, HiCI substantially extends the effective context capacity of LLaMA-2 to 100K (7B) and 64K (13B), and of Qwen3-8B to 48K tokens, and we evaluate it across language modeling, retrieval, and downstream long-context benchmarks. On PG-19 and Proof-pile, HiCI lowers perplexity relative to LongLoRA and ChunkLLaMA across all evaluated context lengths on LLaMA-2, and yields a 14% perplexity reduction on PG-19 and an 8% reduction on Proof-pile for Qwen3-8B at 48K. HiCI further attains 100% passkey retrieval accuracy within the training range and degrades gracefully beyond it. On topic retrieval, HiCI matches GPT-4o-mini-128K and Claude-3.5-Sonnet-200K at perfect (1.00) retrieval accuracy through 13K, and reaches 0.94 at 16K, outperforming all evaluated open-source baselines. On LongBench, HiCI surpasses GPT-3.5-Turbo-16K on both code comprehension (+9.7%) and summarization. Ablations show that both hierarchical stages are necessary, with global integration the larger contributor. Overall, these findings suggest that the construction–integration principle provides an effective inductive bias for long-context modeling. Extending this hierarchical conditioning to vision–language models, where visual tokens significantly increase sequence length, is a promising direction for future work.

## Acknowledgements

This work was supported in part by the Australian Research Council under Projects DP240101848 and FT230100549.

## Impact Statement

This paper presents work whose goal is to advance the field of Machine Learning. There are many potential societal consequences of our work, none which we feel must be specifically highlighted here.

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

# A. Theoretical Analysis

This appendix provides theoretical analysis of HiCI's architectural choices. Rather than establishing optimality, our goal is to characterize the information-theoretic and computational properties that underlie the empirical behaviors observed in experiments: the effectiveness of compact representations, the role of shared compression, and the trade-offs inherent in fixed-capacity hierarchical integration.

## A.1. Notation

Let $X \in \mathbb{R}^{T \times d}$ denote an input sequence of $T$ tokens with hidden dimension $d$. HiCI partitions $X$ into $N = T/S$ non-overlapping segments $\{X_i\}_{i=1}^N$, each of length $S$. Table 5 summarizes the key architectural hyperparameters. All are fixed constants chosen before training and remain invariant across sequence lengths at inference time.

*Table 5.* Summary of notation.

| Symbol | Description |
|---|---|
| $M$ | Local cardinality (queries per segment) |
| $K$ | Global cardinality (context vectors) |
| $d_s$ | Intermediate compression dimension |
| $d_b$ | Bottleneck dimension for attention |

## A.2. Hierarchical Information Flow

We formalize HiCI's hierarchical structure through functional decomposition and analyze the resulting information flow.

**Compositional Structure.** A HiCI block computes the output through three composed functions:

$$L = f_{\text{local}}(X), \qquad G = f_{\text{global}}(L), \qquad \tilde{X} = f_{\text{broadcast}}(X, L, G), \tag{14}$$

where $L = \{L_i\}_{i=1}^N$ with $L_i \in \mathbb{R}^{M \times d}$ denotes the local representations extracted from each segment, and $G \in \mathbb{R}^{K \times d}$ denotes the global context aggregated from all segments. This decomposition directly mirrors the three computational stages described in §3.1.

**Cross-Segment Dependency.** Consider two tokens $x_s \in X_j$ and $x_t \in X_i$ residing in different segments ($j \neq i$). Under standard segmented attention, these tokens cannot interact since attention is restricted within each segment. HiCI overcomes this limitation by introducing a hierarchical pathway:

$$x_s \longrightarrow L_j \longrightarrow G \longrightarrow \tilde{x}_t. \tag{15}$$

This three-hop path enables sequence-wide information flow while preserving the computational benefits of segment-parallel processing.

**Receptive Field.** In the broadcast stage, each token $x_t \in X_i$ attends over the augmented context $[G; L_i; X_i] \in \mathbb{R}^{(K+M+S) \times d}$. The attention output takes the form:

$$\tilde{x}_t = \sum_{j=1}^{K+M+S} \alpha_{tj} \, v_j, \tag{16}$$

where $\{v_j\}$ are value projections and $\{\alpha_{tj}\}$ are softmax-normalized attention weights computed jointly over all $K + M + S$ positions. Since the global context $G$ aggregates information from all $N$ segments, each token gains indirect access to the entire sequence through the first $K$ positions of the augmented context.

## A.3. Causality of Global Context Construction

The Global Integration step in §3.3 aggregates all local representations $\{L_i\}_{i=1}^N$ into a shared global context $G \in \mathbb{R}^{K \times d}$, which is prepended to each segment's key–value sequence. Although token-to-token attention within each segment remains

strictly causal, the shared prefix $G$ introduces a cross-segment non-causal pathway by aggregating representations derived from future segments. To isolate the contribution of this pathway, we replace $G$ with a strictly causal counterpart

$$G_i \;=\; \mathrm{Agg}(L_1, \ldots, L_i),$$

in which segment $i$ aggregates only from current and preceding local representations. All other components of HiCI remain unchanged.

*Table 6.* Perplexity ($\downarrow$) on the PG-19 test set for LLaMA-2-7B at evaluation context lengths of 2K, 4K, 8K. All models are fine-tuned for 1000 steps using HiCI with segment size $S=2048$, and evaluated under standard full attention, consistent with Table 1.

|  | 2K | 4K | 8K |
|---|---|---|---|
| LongLoRA | 7.70 | 7.35 | 7.14 |
| Shared $G$ | 7.27 | 7.01 | 6.93 |
| Causal $G_i$ | 7.28 | 7.00 | 6.94 |

Across all evaluation lengths, $G$ and $G_i$ yield nearly identical perplexity ($|\Delta\mathrm{PPL}| = 0.01$), while both improve over LongLoRA by 0.21–0.43. These results suggest that HiCI's gains are not primarily driven by cross-segment future access, but by the hierarchical Construction–Integration–Broadcast structure.

## A.4. Cardinality Design

The cardinalities $M$ and $K$ govern the capacity of local and global representations, respectively. Here we discuss their design rationale.

**Cognitive Motivation.** Following theories of limited working memory capacity (Miller, 1956; Cowan, 2001), we constrain both $M$ and $K$ to small constants independent of sequence length $T$. This fixed-capacity bottleneck encourages the model to learn hierarchical abstractions rather than relying on token-level memorization.

**Local Cardinality ($M$).** The parameter $M$ determines the number of learnable queries used in local construction, and hence the dimensionality of each local representation $L_i \in \mathbb{R}^{M \times d}$. Empirically, we observe that larger $M$ improves performance at the training context length but degrades generalization to shorter sequences. This behavior is consistent with overfitting to length-specific patterns when excess capacity is available. We set $M = 8$ to balance in-distribution accuracy and length robustness.

**Global Cardinality ($K$).** The parameter $K$ determines the dimensionality of the global context $G \in \mathbb{R}^{K \times d}$. Unlike $M$, the global integration stage operates on a fixed-size input (five statistical summaries), rendering $K$ inherently decoupled from sequence length. We set $K = 4$; the attention-based weighting learns to project the five statistical views into $K$ compact global slots.

## A.5. Local Compression

The local construction stage maps each segment $X_i \in \mathbb{R}^{S \times d}$ to a compact representation $L_i \in \mathbb{R}^{M \times d}$ with $M \ll S$. Motivated by cognitive theories of limited working memory (§3.2), we fix $M$ as a small constant and analyze the information-theoretic implications of this design.

**Capacity Bound.** The cross-attention mechanism projects keys and values into a $d_b$-dimensional subspace before aggregation. Under a standard linear-Gaussian approximation—treating the bottleneck projection as an information channel with effective signal variance $\sigma_X^2$ and noise variance $\sigma_\epsilon^2$—the mutual information between a segment and its local representation admits the capacity-style bound:

$$I(X_i; L_i) \;\lesssim\; M \cdot d_b \cdot \log\left(1 + \frac{\sigma_X^2}{\sigma_\epsilon^2}\right). \tag{17}$$

This bound highlights that the representational budget scales with the product $M \cdot d_b$, not with segment length $S$. While not a tight guarantee for attention in general, it provides a useful characterization of how $(M, d_b)$ jointly control the information throughput of the local interface.

**Inductive Bias.** The fixed bottleneck $(M, d_b)$ forces the model to compress each segment into a small set of salient factors, functioning as an inductive bias toward abstraction. Fine-grained token details must compete for a limited representational budget, favoring task-relevant structure. The capacity–generalization trade-off discussed in §A.4 follows directly from this constraint.

## A.6. Statistical Aggregation

The global integration stage aggregates all local representations $\mathcal{L} \in \mathbb{R}^{(NM) \times d}$ into a fixed-size summary $\mathbf{Z} \in \mathbb{R}^{5 \times d}$ through five complementary statistics. Table 7 describes the information captured by each statistic. Together, these statistics provide

*Table 7.* Statistical summaries computed in global integration.

| Statistic | Captured Information |
|---|---|
| $\boldsymbol{\mu}$ (mean) | Central tendency |
| $\boldsymbol{\sigma}$ (std) | Dispersion |
| $\boldsymbol{\mu}^+, \boldsymbol{\mu}^-$ (max, min) | Extremal activations |
| $\hat{\boldsymbol{\mu}}$ (normalized mean) | Directional structure |

a coarse characterization of the local representation distribution without retaining individual identities.

**Fixed-Size Interface.** A key property of this design is that the intermediate summary $\mathbf{Z} \in \mathbb{R}^{5 \times d}$ remains constant regardless of sequence length $T$ or the number of segments $N$. The subsequent attention-based weighting (§3.3) then projects $\mathbf{Z}$ into the final global context $G \in \mathbb{R}^{K \times d}$ with $K = 4$ slots. This two-stage process decouples global context capacity from sequence length, enabling the same architecture to operate across varying context sizes (see §A.4 for ablations on $K$).

## A.7. Two-Stage Compression

The shared compression $\phi \colon \mathbb{R}^d \to \mathbb{R}^{d_b}$ proceeds through an intermediate bottleneck dimension:

$$\phi = \psi_b \circ \psi_c \colon \quad \mathbb{R}^d \xrightarrow{\psi_c} \mathbb{R}^{d_s} \xrightarrow{\psi_b} \mathbb{R}^{d_b}, \tag{18}$$

with $d_s < d_b \ll d$ ($d_s = 128$, $d_b = 512$, $d = 4096$ in our experiments).

**Regularization via Bottleneck.** The intermediate dimension $d_s$ imposes a capacity constraint before expansion to $d_b$. By the data processing inequality, information in the final representation is bounded by what passes through the narrower bottleneck:

$$I(\mathbf{Z}; \phi(\mathbf{Z})) \leq I(\mathbf{Z}; \psi_c(\mathbf{Z})). \tag{19}$$

This two-stage design forces the model to first identify a compact, task-relevant subspace before expanding to the attention dimension.

**View Invariance.** Applying identical compression parameters to all five statistics enforces *view-invariant* encoding: the model must learn a common projection that preserves relevant information across heterogeneous statistical views. This acts as structural regularization, encouraging consistent representations rather than view-specific overfitting. Our ablations (§4.5) confirm that using separate projections per view yields marginal or no improvement, validating the shared bottleneck design.

## A.8. Computational Complexity

We analyze the computational complexity of HiCI and establish its linear scaling with respect to sequence length.

**Theorem A.1** (Linear Complexity). *HiCI achieves time complexity $O(TSd)$ and space complexity $O(S^2)$ per layer, linear in $T$ for fixed $S$. An additional $O((K+M)d)$ space is required for storing the hierarchical context, which is negligible for typical configurations ($K+M = 12$, $S \geq 1024$).*

*Proof.* Let $N = T/S$ denote the number of segments. We analyze each stage separately.

**Local Construction.** For each segment, cross-attention between $M$ learnable queries and $S$ segment tokens incurs:

$$\underbrace{(M + S) \cdot d \cdot d_b}_{\text{projections}} + \underbrace{M \cdot S \cdot d_b}_{\text{attention}} + \underbrace{M \cdot d_b \cdot d}_{\text{output}} = O(S \cdot d \cdot d_b), \tag{20}$$

where the dominant cost arises from key-value projections over $S$ tokens. Aggregating over $N$ segments yields a total cost of $O(T \cdot d \cdot d_b)$.

**Global Integration.** Computing statistical summaries over all $NM$ local vectors requires $O(NMd) = O(Td/S)$. The subsequent two-stage compression and global attention operate on fixed-size inputs (5 statistics and $K$ queries), contributing $O(1)$ with respect to $T$.

**Top-down Broadcast.** Each segment attends over an augmented context of size $(K + M + S)$:

$$\underbrace{(K + M + S) \cdot d^2}_{\text{projections}} + \underbrace{S \cdot (K + M + S) \cdot d}_{\text{attention}} = O(S^2 \cdot d + S \cdot d^2), \tag{21}$$

where the quadratic dependence on $S$ dominates for typical hidden dimensions. Summing over $N$ segments gives a total cost of $O(T \cdot S \cdot d)$.

**Overall Complexity.** Combining all stages:

$$O(T \cdot d \cdot d_b) + O(T \cdot d/S) + O(T \cdot S \cdot d) = O(T \cdot S \cdot d), \tag{22}$$

where the broadcast stage is asymptotically dominant. For fixed $S$, the overall time complexity is linear in $T$. $\qquad\square$

Table 8 compares HiCI's complexity with related methods. HiCI retains the $O(TSd)$ time complexity of windowed attention while introducing explicit hierarchical cross-segment interactions.

*Table 8.* Computational complexity comparison. $T$: sequence length, $d$: hidden dimension, $S$: segment/window size.

| Method | Time | Space | Cross-Segment |
|--------|------|-------|---------------|
| Standard Attention | $O(T^2 d)$ | $O(T^2)$ | Full |
| Linear Attention | $O(Td^2)$ | $O(Td)$ | Approximated |
| Segmented Attention | $O(TSd)$ | $O(S^2)$ | None |
| LongLoRA | $O(TSd)$ | $O(S^2)$ | Shifted windows |
| **HiCI** | $O(TSd)$ | $O(S^2)$ | Hierarchical |

*Remark* A.2. While HiCI and segmented attention share the same asymptotic complexity, HiCI incurs a small constant overhead from the $(K + M)$ additional context tokens per segment. With $K = 4$ and $M = 8$, this overhead is negligible for typical segment sizes ($S \geq 1024$), adding less than 2% to the attention computation while enabling cross-segment information flow.

# B. Training Details

## B.1. Hyperparameters

Table 9 summarises the hyperparameters used for HiCI training. Unless otherwise specified, the same configuration is adopted for context lengths from 8K to 64K for the 7B model and from 8K to 32K for the 13B model. For the maximum-length settings (100K for 7B and 64K for 13B), we employ DeepSpeed ZeRO Stage-3 and increase the number of segments to $N{=}10$ and $N{=}8$, respectively, to satisfy memory constraints. Supervised fine-tuning on LongAlpaca-12k is performed for 5 epochs; all remaining hyperparameters are held constant.

*Table 9.* Hyperparameters for HiCI training. PT denotes continued pretraining on RedPajama, and SFT denotes supervised fine-tuning on LongAlpaca-12k.

| Hyperparameter | LLaMA-2 | | | Qwen3 |
| --- | --- | --- | --- | --- |
| | 7B (PT) | 13B (PT) | 7B (SFT) | 8B (PT) |
| **Optimization** | | | | |
| Optimizer | AdamW | AdamW | AdamW | AdamW |
| Backbone learning rate | $2 \times 10^{-5}$ | $2 \times 10^{-5}$ | $2 \times 10^{-5}$ | $2 \times 10^{-5}$ |
| HiCI learning rate | $2 \times 10^{-4}$ | $2 \times 10^{-4}$ | $2 \times 10^{-4}$ | $2 \times 10^{-4}$ |
| Weight decay | 0 | 0 | 0 | 0 |
| LR scheduler | Constant w/ warmup | Constant w/ warmup | Constant w/ warmup | Constant w/ warmup |
| Warmup steps | 20 | 20 | 20 | 20 |
| Training duration | 1,000 steps | 1,000 steps | 5 epochs | 1,000 steps |
| **Batch Configuration** | | | | |
| Per-device batch size | 1 | 1 | 1 | 1 |
| Gradient accumulation | 8 | 8 | 8 | 8 |
| Number of GPUs | 8 | 8 | 8 | 8 |
| Effective batch size | 64 | 64 | 64 | 64 |
| **LoRA** | | | | |
| LoRA rank $r$ | 8 | 8 | 8 | 8 |
| LoRA alpha $\alpha$ | 16 | 16 | 16 | 16 |
| LoRA dropout | 0.05 | 0.05 | 0.05 | 0.05 |
| **HiCI Architecture** | | | | |
| Number of segments $N$ | 4 | 4 | 4 | 4 |
| Local slots $M$ | 8 | 8 | 8 | 8 |
| Global slots $K$ | 4 | 4 | 4 | 4 |
| Attention heads | 8 | 10 | 8 | 8 |
| Bottleneck dimension $d_b$ | 512 | 640 | 512 | 512 |
| Compression dimension $d_s$ | 128 | 160 | 128 | 128 |
| Gradient clip (HiCI) | 0.3 | 0.3 | 0.3 | 0.3 |
| **Infrastructure** | | | | |
| Precision | BF16 | BF16 | BF16 | BF16 |
| DeepSpeed | ZeRO Stage-2 | ZeRO Stage-2 | ZeRO Stage-2 | ZeRO Stage-2 |
| Attention kernel | Flash-Attention 2 | Flash-Attention 2 | Flash-Attention 2 | Flash-Attention 2 |

## B.2. Training Efficiency

Figure 3 compares peak GPU memory usage and wall-clock training time for HiCI and LongLoRA across context lengths from 8K to 100K tokens (LLaMA-2-7B, 8×H100-80GB, 1,000 steps; DeepSpeed ZeRO Stage-2 for 8K–64K and Stage-3 for 100K). *In terms of memory*, HiCI introduces a modest overhead of 3.5–9.9% relative to LongLoRA, arising from the learnable local and global representations in the hierarchical pipeline. Since these representations have fixed capacity per segment, the relative memory gap narrows as context length increases and remains manageable even at 100K under ZeRO Stage-3. *In terms of wall-clock time*, while HiCI incurs at most 7.5% additional overhead at short contexts (8K–32K), it becomes progressively faster at long contexts. At 100K tokens, HiCI adopts a finer partitioning with $N{=}10$ segments of 10K tokens each, whereas LongLoRA operates with $N{=}4$ segments of 25K tokens. Because per-segment attention scales quadratically with segment length, this finer-grained grouping substantially reduces the dominant attention compute, yielding a 19.3% reduction in total training time (36.4 h vs. 45.1 h) despite the additional representational overhead.

Table 10 provides a per-layer FLOPs breakdown for Full Attention, $S^2$-Attention, and HiCI across context lengths from 8K to 100K on LLaMA-2-7B. HiCI reduces total FLOPs by 16–60% relative to Full Attention (e.g., 585.0 vs. 996.0 TFLOPs at 32K; 2,762.6 vs. 6,850.7 at 100K), primarily due to segmented attention over partitioned sequences. In addition, the Local Construction and Global Integration (LC+GI) stages introduce only a 1–2% computational overhead compared to $S^2$-Attention (e.g., 585.0 vs. 573.7 at 32K; 2,762.6 vs. 2,727.5 at 100K), reflecting the cost of cross-segment representation construction and aggregation.

*Table 10.* Per-layer FLOPs (TFLOPs) for LLaMA-2-7B across context lengths. LC+GI denotes the combined overhead of the Local Construction and Global Integration stages; dashes indicate stages absent in that method.

| Context | Method | Attn | Proj | FFN | Others | LC+GI | Total |
|---|---|---|---|---|---|---|---|
| 8K | Full Attn | 35.2 | 35.2 | 70.9 | 2.1 | – | 143.4 |
| | S²-Attn | 8.8 | 35.2 | 70.9 | 2.1 | – | 117.0 |
| | HiCI | 9.4 | 35.2 | 70.9 | 2.1 | 2.2 | 119.9 |
| 16K | Full Attn | 140.7 | 70.4 | 141.8 | 4.3 | – | 357.2 |
| | S²-Attn | 35.2 | 70.4 | 141.8 | 4.3 | – | 251.7 |
| | HiCI | 36.4 | 70.4 | 141.8 | 4.3 | 4.4 | 257.3 |
| 32K | Full Attn | 562.9 | 140.7 | 283.7 | 8.6 | – | 996.0 |
| | S²-Attn | 140.7 | 140.7 | 283.7 | 8.6 | – | 573.7 |
| | HiCI | 143.1 | 140.7 | 283.7 | 8.6 | 8.8 | 585.0 |
| 64K | Full Attn | 2251.8 | 281.5 | 567.3 | 17.2 | – | 3117.8 |
| | S²-Attn | 562.9 | 281.5 | 567.3 | 17.2 | – | 1429.0 |
| | HiCI | 567.8 | 281.5 | 567.3 | 17.2 | 17.6 | 1451.4 |
| 100K | Full Attn | 5497.6 | 439.8 | 886.5 | 26.8 | – | 6850.7 |
| | S²-Attn | 1374.4 | 439.8 | 886.5 | 26.8 | – | 2727.5 |
| | HiCI | 1381.9 | 439.8 | 886.5 | 26.8 | 27.6 | 2762.6 |

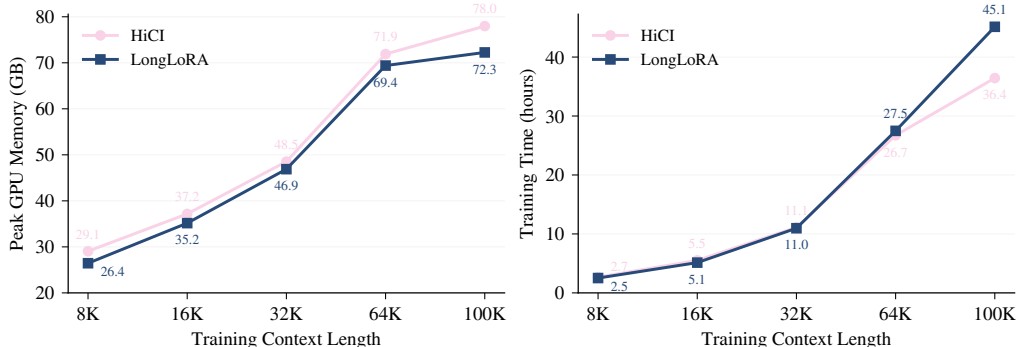

*Figure 3.* Peak GPU memory (left) and wall-clock training time (right) for HiCI and LongLoRA (LLaMA-2-7B, 8×H100-80GB, 1,000 steps; Stage-2 for 8K–64K, Stage-3 for 100K). The three-stage HiCI pipeline raises memory by 3.5–9.9%, which necessitates finer partitioning at long contexts ($N{=}10$ at 100K vs. LongLoRA's $N{=}4$); the resulting quadratic reduction in per-segment attention cost yields a 19.3% wall-clock speedup.

## B.3. Training Loss Trajectories

We compare the training dynamics of HiCI and LongLoRA during LLaMA-2-7B continual pre-training on RedPajama (Computer, 2023) over 2,000 steps, varying context length (8K, 16K) and segment size ($S \in \{1024, 2048\}$). As shown in Figure 4, HiCI with $S{=}1024$ exhibits sustained loss reduction throughout training, with an additional decrease of 38% over the second half at 8K context length and 23% at 16K. In contrast, HiCI with $S{=}2048$ improves only marginally (∼5%), while all LongLoRA variants plateau after approximately 1,000 steps ($\Delta{<}3\%$ thereafter). The two methods display *opposite preferences* with respect to segment granularity. LongLoRA favors coarser segments (final loss 1.69 vs. 1.73 for $S{=}2048$ vs. 1024), consistent with prior findings that shifted sparse attention benefits from wider per-head receptive fields (Chen et al., 2024). In contrast, HiCI improves substantially with finer segmentation (1.01 vs. 1.65), suggesting that a larger number of segments yields richer local representations for hierarchical aggregation. Together, these results indicate that the two cross-segment mechanisms rely on distinct inductive biases: direct attention over wider local windows versus learnable compression and integration over more numerous segments.

## B.4. Parameter Overhead

HiCI introduces additional learnable parameters that are independent of sequence length. Table 11 provides a detailed breakdown for LLaMA-2-7B with 32 transformer layers.

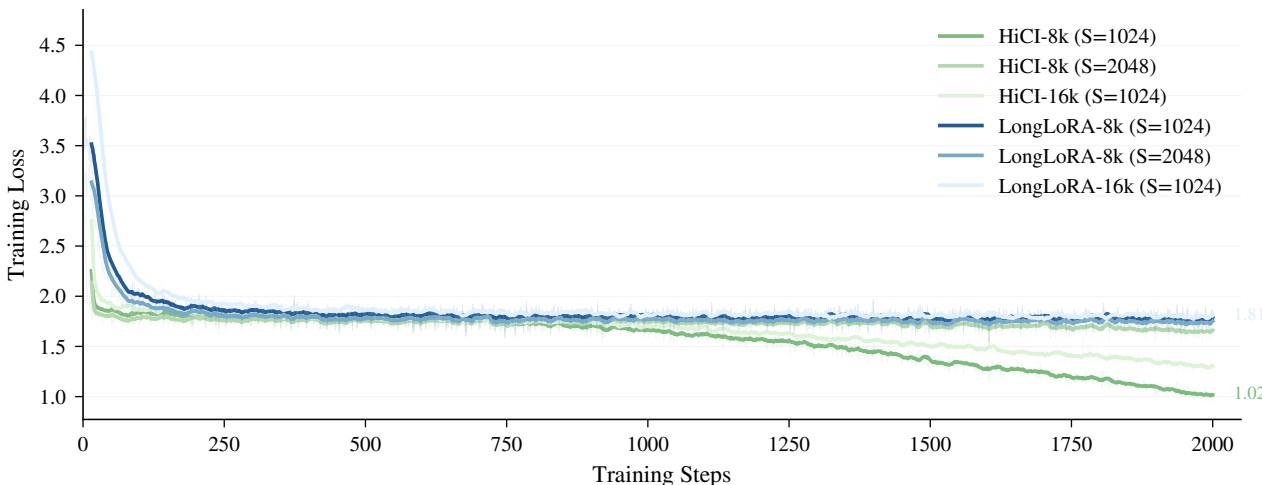

*Figure 4.* Training loss comparison between HiCI and LongLoRA on LLaMA-2-7B continual pre-training (RedPajama, 2,000 steps). Both methods are trained at 8K and 16K context with $S \in \{1024, 2048\}$. HiCI with $S=1024$ sustains optimization throughout, while HiCI with $S=2048$ and all LongLoRA variants plateau beyond step 1,000.

*Table 11.* Parameter overhead of HiCI on LLaMA-2-7B. We use $d=4096$, bottleneck dimension $d_b=512$, shared compression dimension $d_s=128$, $M=8$ local slots, and $K=4$ global slots across 32 layers.

| Module | Component | Per Layer | Total (32L) |
|---|---|---|---|
| *Local Construction* | Memory slots ($M=8$) | 32.8K | 1.0M |
| | Cross-attention (Q/K/V/O) | 8.4M | 268.4M |
| | **Subtotal** | **8.4M** | **269.5M** |
| *Global Integration* | Shared compression ($d_s=128$) | 591.1K | 18.9M |
| | Global queries ($K=4$) | 2.0K | 0.1M |
| | Lightweight attention (Q/K/V/O) | 1.0M | 33.6M |
| | Expansion layer | 2.1M | 67.1M |
| | **Subtotal** | **3.7M** | **119.6M** |
| **HiCI Total** | | **12.2M** | **389.1M** |
| Base Model (LLaMA-2-7B) | | — | 6.74B |
| **Parameter Overhead** | | — | **5.46%** |

The parameter overhead is modest (5.46%) relative to the base model and, importantly, does not scale with sequence length—the same parameters handle 4K, 32K, or 100K contexts without modification.

## C. Layer-wise Attention Analysis

We analyze how HiCI routes attention across hierarchical representations by recording layer-wise attention statistics during evaluation on PG-19. For each layer, we compute the fraction of total attention mass assigned to the $K=4$ global slots, averaged over all heads and evaluation samples. At each layer, the key–value sequence consists of $K$ global slots, $M=8$ local slots, and $S$ segment tokens, yielding a total length of $K+M+S$ (1036 for $S=1024$; 2060 for $S=2048$). Fig. 5(a) compares $S=1024$ and $S=2048$ under matched conditions (8K evaluation, 2K training steps), while Fig. 5(b) probes robustness at $S=2048$ by varying evaluation length and training duration.

**Depth-dependent routing.** Across configurations, attention to global slots exhibits a clear increasing trend with layer depth, despite minor layer-wise variations. Averaged over early layers (L0–7), global attention ranges from 1% to 8% across configurations; for deep layers (L24–31), it ranges from 6% to 26%, yielding deep-to-early ratios of 3.3–4.9×. At the final layer (L31), global attention reaches 40.4% for $S=1024$ ($\approx 105\times$ the uniform baseline of 0.39%) and 12.7% for $S=2048$ ($\approx 65\times$ the baseline of 0.19%). As no explicit supervision is imposed on attention allocation, this stratification suggests that

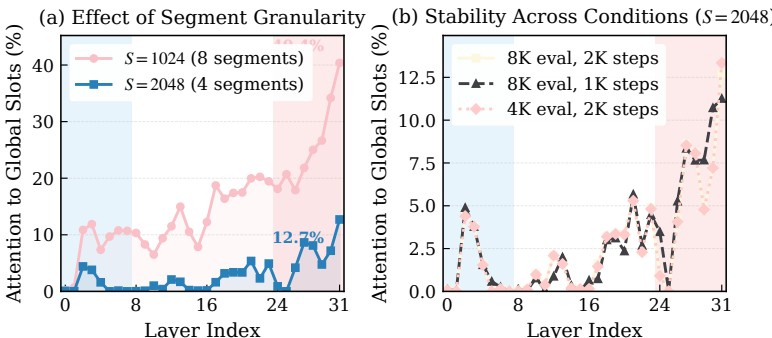

*Figure 5.* Layer-wise attention allocated to global slots during evaluation on PG-19. Background shading denotes depth groups: early (L0–7, blue), middle (L8–23, white), and deep (L24–31, red). **(a)** Comparison of segment sizes $S$=1024 and $S$=2048 under matched conditions (8K evaluation, 2K steps): finer segmentation yields substantially higher attention to global slots, with the final layer (L31) reaching 40.4% versus 12.7%. **(b)** Robustness at $S$=2048: varying evaluation length (8K vs. 4K) and training steps (2K vs. 1K) yields nearly identical layer-wise allocation patterns, with per-layer deviations within 1 percentage point. In both panels, attention to global slots increases toward deeper layers.

deeper layers allocate attention preferentially to hierarchical context.

**Effect of segment granularity.** Reducing $S$ from 2048 to 1024 amplifies global attention by approximately 4–7× across all depth groups, substantially exceeding the 2× change in the proportional presence of global slots ($\frac{4}{1036}$ vs. $\frac{4}{2060}$). This behavior is consistent with the divergent scaling observed in Section 4.5: finer segmentation tightens the local information bottleneck and is associated with stronger reliance on global aggregation.

**Robustness.** At fixed $S$=2048, layer-wise allocation patterns remain largely stable when the evaluation length is halved (8K→4K) or when training is shortened (2K→1K steps), with per-layer deviations within 1 percentage point for most layers. This suggests that hierarchical routing emerges early in training and generalizes across sequence lengths.

