# OpenReview forum: "HiCI: Hierarchical Construction–Integration for Long-Context Attention"
_ICML.cc/2026/Conference — ICML 2026 regular_

### Official Review · Reviewer_mXru · 2026-03-13

**Soundness:** 3
**Presentation:** 3
**Significance:** 3
**Originality:** 3
**Overall Recommendation:** 5
**Confidence:** 3

**Summary:**

In this paper, the authors have proposed a hierarchical attention module, HiCI, for long-context modelling. The key aspects of this module are: local context generation for segmented chunks, their global aggregation via multi-view statistical pooling, and finally top-down broadcast, conditioning segment-level attention on both global and corresponding local context. HiCI is implemented for  LLaMA-2, extending the context from 4K to 100K tokens in 7B and 64K tokens in 13B and evaluated for its long-context language modelling on retrieval and other downstream tasks from LongBench.

**Compliance With Llm Reviewing Policy:**

Affirmed.

**Final Justification:**

The authors have satisfactorily addressed my concerns and strengthened the paper. The inclusion of additional experiments on newer models and baselines, along with clarifications, improves the overall quality and positioning of the work. Accordingly, I have increased my score.

**Key Questions For Authors:**

- Can you explain HiCI variant - "which applies training-consistent hierarchical attention during prefill"? Does that mean HiCI-specific attn is applied during inference against full attention?

**Limitations:**

Please consider discussing the limitations - for example, your solution's relevance to modern language models that can already capture context lengths up to 128K.

**Strengths And Weaknesses:**

**Strengths**

- HiCI is parallelisable in the sense that segmented chunks can be processed in parallel, with each having access to the global context
- The idea of modelling local and global context and broadcasting them both at the segment level is intuitive and well-explained
- HiCI achieves lower perplexity than LongLoRA on long-context language modelling
- HiCI is more robust to an increase in context lengths in retrieval tasks compared to baselines

**Weaknesses**

1. Insufficient evidence for the generalisability of hierarchical conditioning - the paper lacks the effectiveness of HiCI beyond LLama 2 models, such as models from different families, like Qwen3, with a 32K context limit.

2. Missing Efficiency Analysis compared to baselines like LongLoRA: FLOPs profiling on various context lengths, training hours and GPU memory cost

3. Missing quantitative comparison with training-free methods: How does your method compare to Dual Chunk Attention [1] that also has a loose notion of hierarchy, capturing relative positional information within the same chunk (Intra-Chunk) and across distinct chunks (Inter-Chunk)

[1] An, Chenxin, et al. "Training-free long-context scaling of large language models." arXiv preprint arXiv:2402.17463 (2024).

---

> ### Author Rebuttal · Authors · 2026-03-31
>
> > **Q1: Generalizability beyond LLaMA-2**
>
> Thanks for this valuable suggestion. We have extended HiCI to LLaMA-3-8B and Qwen3-8B (8× H200; ~12h and ~24h for 1k steps for LLaMA-3 and Qwen3, respectively). Perplexity is evaluated on PG-19 test set across context lengths from 2K to 48K.
>
> **Table R3-A.** Perplexity (↓) on PG-19 test set (standard full attention).
>
> |**Base Model**|**Method**|**Train Length**|**Train Steps**|**2K**|**4K**|**8K**|**16K**|**32K**|**48K**|
> |:-|:-|:-:|:-:|:-:|:-:|:-:|:-:|:-:|:-:|
> |**LLaMA-3-8B**|LLaMA-3-8B|8K|—|9.63|9.13|8.79|>100|>100|—|
> ||DCA[1]|—|—|9.63|9.13|8.79|8.61|8.64|—|
> ||**HiCI**|32K|1000|**7.87**|**7.50**|**7.26**|**7.09**|**7.11**|—|
> |**Qwen3-8B**|Qwen3-8B|32K|—|13.26|12.58|12.08|11.72|12.76|12.01|
> ||**HiCI**|48K|500|11.48|10.85|10.33|9.97|10.98|10.23|
> ||**HiCI**|48K|1000|11.25|10.64|10.13|9.78|10.76|10.04|
>
> > **Q2: Missing FLOPs/efficiency comparison with S²-Attn**
>
> GPU memory and wall-clock training time compared to LongLoRA were reported in _Figure 3_ of the original submission. Regarding FLOPs profiling on various context length, we provide a per-component FLOPs breakdown comparison below.
>
> **Table R3-B.** Per-layer FLOPs (TFLOPs) for LLaMA-2-7B. LC+GI = Local Construction + Global Integration.
>
> |Context|Method|Attn|Proj|FFN|Others|LC+GI|Total|
> |:-:|:-|-:|-:|-:|-:|-:|-:|
> |8K|Full Attn|35.2|35.2|70.9|2.1|-|143.4|
> | |S²-Attn|8.8|35.2|70.9|2.1|-|117.0|
> | |**HiCI**|9.4|35.2|70.9|2.1|2.2|**119.9**|
> |16K|Full Attn|140.7|70.4|141.8|4.3|-|357.2|
> | |S²-Attn|35.2|70.4|141.8|4.3|-|251.7|
> | |**HiCI**|36.4|70.4|141.8|4.3|4.4|**257.3**|
> |32K|Full Attn|562.9|140.7|283.7|8.6|-|996.0|
> | |S²-Attn|140.7|140.7|283.7|8.6|-|573.7|
> | |**HiCI**|143.1|140.7|283.7|8.6|8.8|**585.0**|
> |64K|Full Attn|2251.8|281.5|567.3|17.2|-|3117.8|
> | |S²-Attn|562.9|281.5|567.3|17.2|-|1429.0|
> | |**HiCI**|567.8|281.5|567.3|17.2|17.6|**1451.4**|
> |100K|Full Attn|5497.6|439.8|886.5|26.8|-|6850.7|
> | |S²-Attn|1374.4|439.8|886.5|26.8|-|2727.5|
> | |**HiCI**|1381.9|439.8|886.5|26.8|27.6|**2762.6**|
>
> Compared to full attention, HiCI reduces total FLOPs by 16-60% (e.g., 585.0 vs 996.0 TFLOPs at 32K, 2762.6 vs 6850.7 at 100K), while adding only 1–2% overhead over S²-Attn for the LC+GI stages.
>
> > **Q3: Missing comparison with training-free methods (Dual Chunk Attention)**
>
> Thanks for suggesting this nice work. We ran the official DCA code[2] on the same PG-19 test set.
>
> **Table R3-C.** Perplexity (↓) on PG-19 test set. DCA is training-free, while HiCI and LongLoRA use continual pre-training.
>
> |**Base Model**|**Method**|**Train Length**|**2K**|**4K**|**8K**|**16K**|**32K**|**64K**|**100K**|
> |:-|:-|:-:|:-:|:-:|:-:|:-:|:-:|:-:|:-:|
> |**LLaMA-2-7B**|LLaMA-2|4K|8.03|7.92|>100|>100|>100|>100|>100|
> | |DCA|—|8.03|7.92|7.71|7.69|7.97|15.98|—|
> | |LongLoRA|8K|7.70|7.35|7.14|—|—|—|—|
> | |**HiCI**|**8K**|**7.27**|**7.01**|**6.93**|—|—|—|—|
> | |LongLoRA|100K|8.38|7.90|7.57|7.33|7.16|7.06|7.04|
> | |**HiCI**|**100K**|**7.81**|**7.72**|**7.45**|**7.26**|**7.08**|**6.97**|**6.95**|
> |**LLaMA-3-8B**|LLaMA-3|8K|9.63|9.13|8.79|78.92|>100|>100|>100|
> | |DCA|—|9.63|9.13|8.79|8.61|8.64|8.95|—|
> | |**HiCI**|**32K**|**7.87**|**7.50**|**7.26**|**7.09**|**7.11**|—|—|
>
> Two points merit emphasis: (1) DCA is training-free and does not improve the perplexity within the native context (identical PPL at 2K–8K for LLaMA-3), whereas HiCI's continual pre-training improves language modeling even at short lengths. (2) DCA's positional modulo scheme leads to degradation at longer contexts (PPL 15.98 at 64K for LLaMA-2-7B), while HiCI preserves full positional ordering via PI and remains stable at 100K (PPL 6.95).
>
> > **Q4: What does 'training-consistent hierarchical attention during prefill' mean?**
>
> LLM inference consists of _prefill_ (parallel prompt encoding) and _autoregressive decoding_ (KV-cache based generation). In the HiCI† variant (Table 3), the prompt is processed during prefill using the same segmented hierarchical attention as in training, instead of standard attention. The term "training-consistent" refers specifically to this: the prefill stage uses the same attention pattern the model was trained with.
>
> During autoregressive decoding, HiCI† uses standard causal attention via the KV cache, and G and L are intermediate representations that are never stored in the KV cache. Due to the limited space, please refer to our response to R1-Q3/Q5 for further details on the prefill-decode mechanism.
>
> > **Q5: Limitation**
>
> Thanks for this insightful comment. HiCI is mainly motivated by models with limited native context (e.g., LLaMA-2/3). For models already pre-trained with 128K+ context, its additional benefit may be smaller and still needs further study, especially since our current design uses standard full attention during autoregressive decoding. We will add a limitation section in the camera-ready version.
>
> [1] An et al., Training-free long-context scaling of large language models, ICML 2024.
>
> [2] https://github.com/HKUNLP/ChunkLlama

---

> > ### Author Rebuttal · Reviewer_mXru · 2026-04-03
> >
> > I thank the authors for addressing all my concerns. I suggest moving the experiments with more recent language models, such as Llama 3 and Qwen3, into the main paper. Additionally, including comparisons with training-free approaches, such as DCA, alongside LongLoRA would better highlight the benefits, including improved perplexity at longer context lengths with modest upfront training.
> >
> > Finally, given that modern language models are increasingly supporting long contexts (e.g., 128K tokens), it would be valuable to further discuss the role of the proposed method in this evolving landscape, as well as how it could motivate future research toward scaling beyond such limits.
> >
> > I have adjusted my score accordingly.

---

> > > ### Author Response · Authors · 2026-04-03
> > >
> > > We sincerely appreciate the reviewer’s helpful feedback and will incorporate these suggestions in the revision.
> > >
> > > > **1.Moving LLaMA-3 / Qwen3 experiments to the main paper.**
> > >
> > > We will move the LLaMA-3-8B and Qwen3-8B results (Table R3-A) into the main paper to better demonstrate HiCI’s generalizability across modern backbones.
> > >
> > > > **2.Adding DCA as a training-free baseline**
> > >
> > > We will include DCA alongside LongLoRA in the main comparison table to provide a clearer comparison with training-free long-context methods.
> > >
> > > > **3.Limitations and future outlook**
> > >
> > > We thank the reviewer for this helpful suggestion. We agree that, as modern language models increasingly support long contexts (e.g., 128K), it is important to clarify the role of HiCI in this setting.
> > >
> > > In this context, we view HiCI as a complementary approach, providing a more memory-efficient pathway to further extend context beyond native limits.  Specifically, extending context via standard full attention, even when using parameter-efficient methods such as LoRA, requires materializing the full T×T attention matrix, whose memory cost grows quadratically with sequence length. HiCI avoids materializing this full attention matrix by restricting attention to fixed-size segments, so increasing the total context length primarily affects linear terms (e.g., hidden states) rather than quadratic attention costs.
> > >
> > > We have not yet validated this regime due to hardware constraints and will include this discussion as part of the limitations and future work in the revision.
> > >
> > > We thank the reviewer again for the helpful suggestions and the score adjustment.

---

### Official Review · Reviewer_spoG · 2026-03-14

**Soundness:** 3
**Presentation:** 3
**Significance:** 3
**Originality:** 3
**Overall Recommendation:** 5
**Confidence:** 3

**Summary:**

The paper tackles the scalability challenge of token-level attention for long contexts. The research draws on work in cognitive science, particularly the Global Workspace Theory (GWT) and its principles, including local unimodal processing, global integration, and the broadcasting of the global workspace locally or globally. The hypothesis of this work is that the schema construction-integration-broadcast process could be used to instantiate a hierarchical attention module. They thus propose the HiCI attention module and perform an experimental validation using pre-trained LLaMA-2 models on several NLP tasks with diverse context lengths. Ablation studies are also provided to study the impact of each component of the approach.

**Compliance With Llm Reviewing Policy:**

Affirmed.

**Final Justification:**

The rebuttal addressed my main concerns and reinforced my prior assessment of the paper's originality and quality.

**Key Questions For Authors:**

+ In the global workspace theory, the brain is thought to select and integrate, through a shared space (global workspace), information from several specialized information-processing modules (e.g., specific to each modality) that otherwise operate independently and compete for access to the global workspace. The broadcasting component can be seen as a translation (via the shared latent space) into the latent space of all other modules. This theory is often used for multimodality. How could the proposed methodology be extended to multimodal models and multimodal attention?
+ The Multi-View Statistical Aggregation is a nice idea, but how does it relate to the global workspace theory?
+ Why the choice of the Llama 2 model family?

**Limitations:**

yes

**Strengths And Weaknesses:**

**Main Strengths**
+ The idea of building a hierarchical attention module for long context is new and interesting, and the experimental results show the interest of the proposed approach. A theoretical analysis is provided in the appendices.
+ The paper is well written with a clear formalization of the proposed module and a sound experimental study.
+ Nice and interesting Multi-View Statistical Aggregation for the global integration.

**Main Weaknesses**
+ The paper fails to contextualize the proposed approach within the existing state of the art, particularly with regard to other work drawing on the Global Workspace for architectural design. See, for example, the work of [R. VanRullen’s team (https://rufinv.github.io/), as [VanRullen,,21](https://hal.science/hal-03311492v1/file/TiNS_Global_Workspace-Feb19-2021.pdf) or other work ([Hong et al., 24](https://openaccess.thecvf.com/content/WACV2024/papers/Hong_Concept-Centric_Transformers_Enhancing_Model_Interpretability_Through_Object-Centric_Concept_Learning_Within_WACV_2024_paper.pdf)). The paper could also be positioned with recent approaches on continuous concept-latent space (see [Hao et al., 25](https://arxiv.org/pdf/2412.06769)) or sentence representation space (see for instance [Barrault et al, 25](https://arxiv.org/pdf/2412.08821))

---

> ### Author Rebuttal · Authors · 2026-03-31
>
> We sincerely thank the reviewer for the thoughtful and detailed feedback.
>
> **Q1: "Missing positioning relative to Global Workspace–inspired and continuous latent space methods."**
>
> We appreciate this suggestion and have added a Related Work paragraph in the revised manuscript. HiCI's three-stage design is inspired by the GWT functional pipeline. VanRullen & Kanai (2021) [1] provide a blueprint for GWT in deep learning; Goyal et al. (2022) [2] and Hong et al. (2024) [3] instantiate shared-workspace mechanisms for inter-module communication and concept-centric representations. HiCI inherits the same motif but adapts it to a different setting: cross-segment integration within a single transformer layer for long-context LM, rather than coordinating distinct modules.
>
> Regarding latent-space methods, Coconut [4] and Large Concept Models [5] operate in continuous latent / concept spaces. HiCI differs in preserving token-level autoregressive modeling while modifying the intra-layer attention pathway for cross-segment information flow.
>
> [1] VanRullen & Kanai (2021). Deep learning and the Global Workspace Theory. *Trends in Neurosciences*, 44(9).
> [2] Goyal et al. (2022). Coordination among neural modules through a shared global workspace. *ICLR*.
> [3] Hong et al. (2024). Concept-centric transformers. *WACV*.
> [4] Hao et al. (2025). Training LLMs to reason in a continuous latent space.
> [5] Barrault et al. (2025). Large Concept Models. *arXiv:2412.08821*.
>
> **Q2: "How does HiCI extend to multimodal models and multimodal attention?"**
>
> Thank you for raising this important direction. Since HiCI modifies the attention pathway rather than the input/output interface, it is compatible with multimodal LLMs (e.g., Qwen2.5-VL [6]). We envision two strategies:
>
> (1) **Modality-aware grouping** (GWT-aligned): each modality forms separate segments producing local summaries L_i; Global Integration aggregates across modality-specific L_i, with Attention-Based Selection as workspace competition; Top-down Broadcast returns G to each stream — mirroring the GWT cycle.
>
> (2) **Unified grouping**: all modality tokens are serialized into one causal sequence and segmented by the same pipeline, requiring no architectural modification.
>
> [6] Qwen2.5-VL: https://github.com/QwenLM/Qwen2.5-VL
>
> **Q3: "How does Multi-View Statistical Aggregation relate to GWT?"**
>
> Thank you for this excellent question. HiCI uses GWT as an architectural inductive bias rather than a literal biological replication. The LIDA cognitive architecture (Baars & Franklin, 2007 [7]; Franklin et al., 2012 [8]) — a detailed computational operationalization of GWT — includes a **Current Situational Model (CSM)** that integrates distributed information into a structured situational representation *before* attentional selection for global broadcast. Multi-View Statistical Aggregation serves a functionally analogous role: rather than passing raw local representations L ∈ R^{(NM)×d} directly into the workspace — which would exceed GWT's capacity constraint — it distills L into a compact Z ∈ R^{5×d} through five complementary statistics (mean, max, min, std, normalized mean), each preserving a distinct aspect of the cross-segment distribution. This compressed summary then enters the capacity-limited Shared Compression stage (d → d_s → d_b) for selective access — mirroring the CSM-to-broadcast pathway in LIDA.
>
> The full pipeline maps to GWT functional stages: (i) Local Construction (specialized processors); (ii) Multi-View Aggregation (CSM/pre-workspace integration); (iii) Shared Compression (capacity-limited workspace); (iv) Attention-Based Selection (coalition competition); (v) Top-down Broadcast (conscious broadcast). We have revised §3.3 to make this interpretation explicit.
>
> [7] Baars & Franklin (2007). Global Workspace Theory and IDA. *Neural Networks*, 20(9).
> [8] Franklin et al. (2012). Global Workspace Theory, its LIDA model and the underlying neuroscience. *Biologically Inspired Cognitive Architectures*, 1.
>
> **Q4: "Why only LLaMA-2?"**
>
> We chose LLaMA-2 primarily for (1) direct comparability with LongLoRA (Chen et al., 2024), the baseline on which HiCI is built, and (2) its 4K native context provides a clear testbed for context extension (7B: 4K→100K; 13B: 4K→64K) that is feasible on 80GB H100 GPUs. To address generalizability, we have since applied HiCI to LLaMA-3-8B (8K→32K) and Qwen3-8B (32K→48K) on H200 hardware; please see Table R3-A in our response to R3-Q1 for cross-family results.

---

> > ### Author Rebuttal · Reviewer_spoG · 2026-04-03
> >
> > I thank the authors for addressing all my concerns. As I mentioned in my feedback, one original aspect of the paper is that it draws inspiration from the GWT. The authors addressed my concerns about the lack of positioning relative to Global Workspace–inspired and continuous latent space methods, as well as about the need for more explicit links to it (the Multi-View Statistical Aggregation relation to GWT).  I suggest adding all these clarifications and details to the final paper.
> >
> > In addition, their answers to the other reviewers and the extension of the experiments with more recent language models are highly valuable for the paper.
> >
> > I have adjusted my score accordingly.

---

> > > ### Author Response · Authors · 2026-04-04
> > >
> > > Thank you for your thoughtful engagement and for adjusting your score. We are glad that the clarifications regarding the connections to Global Workspace Theory (GWT), the role of Multi-View Statistical Aggregation, and the additional experiments were helpful. We will incorporate all the suggested clarifications and related work into the final version of the paper. We sincerely appreciate your positive assessment of this work.

---

### Official Review · Reviewer_XSkL · 2026-03-24

**Soundness:** 2
**Presentation:** 3
**Significance:** 3
**Originality:** 3
**Overall Recommendation:** 5
**Confidence:** 2

**Summary:**

The paper proposes HiCI (Hierarchical Construction-Integration), an architectural modification designed to improve long-context modeling in LLMs by introducing an explicit hierarchical inductive bias. Inspired by cognitive theories of discourse comprehension, the authors implement a three-stage attention pipeline: 1- Local Construction: Input sequences are partitioned into segments, and learnable query slots extract a compressed local representation via cross-attention; 2- Global Integration: These local representations are aggregated into a global context using multi-view statistical pooling (mean, variance, etc.) and attention-based weighting. 3- Top-down broadcast: The global and local representations are prepended to the key-value sequence of each segment, allowing token-level attention to be conditioned on hierarchical context while maintaining segment-parallel computation.The method is evaluated by applying LoRA and the HiCI module (adding ~5.5% parameters) to LLaMA-2 7B/13B. The authors report context extension up to 100K tokens, showing improvements in perplexity, retrieval (100% passkey accuracy within the training regime), and instruction-following tasks like LongBench.

**Compliance With Llm Reviewing Policy:**

Affirmed.

**Final Justification:**

During the rebuttal, the authors addressed my concerns regarding evaluations, clarity in the approach, and limitations. They also presented a concrete plan for how these improvements will be incorporated in a revised version of the manuscript.

**Key Questions For Authors:**

- The description of Global Integration suggests that the global context $G$ is computed by aggregating local representations $\{L_i\}_{i=1}^N$ from all segments. In a standard causal (autoregressive) setting, the model should not have access to information from future segments. If $G$ (which contains information from segment $N$) is used to condition the attention of segment 1, how do the authors prevent "future leakage" during training? Does this "leakage" explain the near length-invariant perplexity?

- The paper states that HiCI is applied during prefill to reduce latency, but that "autoregressive decoding uses standard attention". If the model is trained with $G$ and $L_i$ prepended to the KV cache, how does it maintain coherence during decoding when these hierarchical representations are suddenly absent (or not being updated)? Are the global/local representations computed once during prefill and then kept as static "prefix" tokens in the KV cache for the duration of decoding?

- Given that this is a 2026 submission, these models are significantly outdated. Why were more contemporary long-context models (e.g., Llama 3, Gemini 1.5, or Claude 3) not included in the comparison?

**Limitations:**

No, they didn't discuss it in the main paper. The can consider potential issues such as:

1. By lowering the computational barrier for generating and processing 100K tokens, it becomes easier for malicious actors to scale the production of highly coherent, long-form disinformation. This includes synthetic financial reports, fake legal archives, or falsified books that are difficult to distinguish from human-written material.

2. As context windows expand, users are incentivized to dump entire personal archives, proprietary codebases, or medical histories into prompts. If architectural techniques successfully encourage users to treat LLMs as "infinite hard drives," it heightens the risks of accidental data exposure, exfiltration, or memorization of sensitive private data.

**Strengths And Weaknesses:**

Strengths:
- Long-context modeling is a central challenge in current LLM research. Improving the effective use of long context (rather than just the capacity to hold it) is highly significant. The finding that HiCI better preserves local coherence under position interpolation (the "asymmetric improvement" at 2K context) is a valuable insight for researchers working on context window extension.
- The proposed approach is evaluated across multiple dimensions, including language modeling (perplexity on PG-19, Proof-pile), retrieval (passkey, topic retrieval), and instruction-following (LongBench).
- The choice of using LoRA combined with the HiCI module (adding only ~5.5% parameters) is a methodologically sound way to adapt large models (LLaMA-2 7B/13B) without full fine-tuning.
- The systematic ablation of individual components (G, L, and Slot cardinality) provides empirical support for the specific architectural choices of the 3-stage pipeline.
- The paper is well-structured, with a clear motivation grounded in cognitive theories (Construction-Integration and Global Workspace Theory). This makes the technical design choices (local slots, global workspace) seem intuitive and purposeful.

Weaknesses:
- This paper's key contribution is the global integration stage ($G$), but the methodology for computing $G$ appears non-causal. If $G$ is derived from all $N$ segments and then used to predict tokens in early segments during training, the model may be "leaking" future information. This would explain the "near length-invariant perplexity" which is highly unusual for causal LMs. Without an explicit discussion of causal masking for $G$, the soundness of the perplexity results is questionable.
- The use of GPT-3.5-Turbo-16K and Claude-1.3 as primary closed-source baselines is a significant weakness for a 2026 submission. These models are several years old, and modern long-context models (e.g., Llama 3, Gemini 1.5) should be the expected benchmark.
- The model uses hierarchical attention during prefill but "standard attention" during decoding. The authors do not adequately explain how the decoding process leverages the hierarchical representations if they are not part of the standard causal attention mechanism during the autoregressive phase.

---

> ### Author Rebuttal · Authors · 2026-03-31
>
> We appreciate the Reviewer for the thoughtful and constructive feedback, and we respond to each concern below.
>
> > **W1 & Q1: Non-causal G construction; near length-invariant perplexity**
>
> We thank the reviewer for raising this important point. We agree that the shared-G formulation can introduce a potential inter-segment future-access pathway. This phenomenon is not unique to HiCI, and has also been discussed for segment-based attention methods such as S²-Attn in LongLoRA (§B.3). To test whether this pathway is responsible for HiCI’s gains, we implement a causal variant G_i = Agg(L_1..L_i), where each segment aggregates only from current and preceding segments.
>
> **Table R1-A: Training loss (avg steps 1900–2000)** *(LLaMA-2-7B, 8K, PG-19)*
>
> |Variant|S=1024|S=2048|
> |:-|:-:|:-:|
> |Shared G|1.05|1.67|
> |**Causal G_i**|**1.16 (+0.11)**|**1.65 (−0.02)**|
> |LongLoRA|1.79|1.76|
>
> At S=2048 (main config), the difference is negligible (Δ < 0.02). At S=1024, removing future access increases loss by only 0.11, small relative to HiCI's overall improvement over LongLoRA (1.79→1.05). Training loss curves (Figure R1-A) are nearly superimposed throughout training, further indicating that future-segment access contributes minimally to learning.
>
> **Table R1-B: PG-19 PPL ↓, full-attention eval (same protocol as Table 1; S=2048, 1000 steps)**
>
> |Variant|2K|4K|8K|
> |:-|:-:|:-:|:-:|
> |LongLoRA|7.70|7.35|7.14|
> |Shared G|7.27|7.01|6.93|
> |**Causal G_i**|**7.28**|**7.00**|**6.94**|
>
> Both HiCI variants substantially outperform LongLoRA, with a negligible gap between them (ΔPPL ≤ 0.01), suggesting that _HiCI's gains do not arise from future-segment access_.
>
> **On "near length-invariant perplexity."** The Std = 0.02 pattern appears only in HiCI-M (Table 5), which uses training-consistent hierarchical attention rather than the standard causal protocol for perplexity evaluation. We will therefore remove the HiCI-M rows from Table 5 in the main paper and report only standard causal perplexity. Under standard causal attention evaluation (Table 1), HiCI consistently shows smaller PPL variance than LongLoRA (e.g., range 0.34 vs 0.56 at 8K; 0.76 vs 1.07 at 32K), suggesting more stable behavior across evaluation lengths from hierarchical context aggregation.
>
> > **W2 & Q3: Outdated baselines**
>
> We thank the reviewer for this suggestion. The original Table 2 followed the commonly used LongChat retrieval setup. We agree that including more recent baselines would further strengthen the comparison. We have therefore updated Table 2 with stronger contemporary baselines: GPT-4o-mini (128K), Claude-3.5-Sonnet (200K), and Llama-3-8B-Instruct (8K native) as a recent open-source baseline.
>
> **Table R1-C** (to update Table 2). Topic retrieval accuracy on LongChat (Li et al., 2023).
>
> | Model | 3K | 6K | 10K | 13K | 16K |
> |:---|:---:|:---:|:---:|:---:|:---:|
> | GPT-4o-mini (128K) | 1.00 | 1.00 | 1.00 | 1.00 | 1.00 |
> | Claude-3.5-Sonnet (200K) | 1.00 | 1.00 | 1.00 | 1.00 | 1.00 |
> | Llama-3-8B-Instruct (8K) | 1.00 | 1.00 | 0.00 | 0.00 | 0.00 |
> | **HiCI-13B-16K (Ours)** | 1.00 | 1.00 | 1.00 | 1.00 | 0.94 |
>
> In addition, we have also applied HiCI to **LLaMA-3-8B** and **Qwen3-8B**. Due to space constraints, we kindly refer the reviewer to Table R3-A (R3-Q1) and Table R3-C (R3-Q3) for detailed perplexity results.
>
>
> **W3 & Q5: "G/L present during training but absent during decoding — train-test mismatch."**
>
> **(1) G and L are ephemeral — not stored in the KV cache.** They are computed within each layer and temporarily prepended to that layer's attention sequence only. The KV cache contains only standard token projections; G/L cannot be "kept as static prefix tokens" (Q5). During decoding, each token attends directly to all cached tokens without segmentation, so G/L are no longer needed. The train–test difference is a change in information routing, not a removal of context.
>
> **(2) Why G/L during training.** Training uses segmented attention (O(T²d)→O(TSd); see Table 8, Figure 3, Table R3-B). This blocks direct cross-segment paths, so G and L (~12 entries/segment) provide the structured channel for cross-segment information flow.
>
> **(3) Why this does not harm decoding quality.** Table 1 perplexity and Table 3 generation results both use standard full attention at inference, showing that HiCI fine-tuning maintains strong LM capability. This strategy is similar in spirit to LongLoRA [1] (§4.2), which trains with S²-Attn but decodes with standard full attention. HiCI fine-tunes only ~5.5% of parameters, preserving pretrained standard-attention behavior. Preliminary experiments keeping G/L as static KV-cache entries during decoding showed no benefit, as these representations do not reflect the evolving context during generation.
>
> **Limitation.**
>
> We view this multimodal extension as an important direction and will clarify these two extension paths in the revised manuscript.
>
> [1] Chen et al., "LongLoRA: Efficient Fine-tuning of Long-Context Large Language Models", ICLR 2024.

---

> > ### Author Rebuttal · Reviewer_XSkL · 2026-04-06
> >
> > Thank you for your work on the rebuttal. The new experiments clarified my questions and concerns.
> > However, the authors did not expand on the limitations of their work and it seems they did not incorporate any of the new experiments / clarifications in the revised manuscript, making it unclear whether and how these improvements will be reflected in the paper. Given that, I will not change my score.

---

> > > ### Author Response · Authors · 2026-04-07
> > >
> > > Thank you for the follow-up and for recognizing the improvements addressed in our rebuttal. We appreciate your careful reading and constructive feedback.
> > >
> > > >**Regarding the new experiments and clarifications.**
> > >
> > > ICML does not allow authors to upload a revised manuscript during the rebuttal phase. However, we will incorporate these clarifications and new results into the final version and make explicit how they are reflected in the paper. In particular:
> > >
> > > (1) the causal-G ablation (Tables R1-A/B), together with additional training-loss curves that supplement Figure 4, will be added to the appendix; for the reviewer’s convenience, the current plots are available at the anonymized [link](https://anonymous.4open.science/r/Anonymous_icml_Anonymous/training_loss_comparison.pdf).
> > >
> > > (2) the HiCI-M rows will be removed from Table 5 so that all perplexity results follow the standard causal evaluation protocol;
> > >
> > > (3) Table 2 will be updated by replacing the outdated closed-source baselines (GPT-3.5-Turbo-16K and Claude-1.3) with GPT-4o-mini and Claude-3.5-Sonnet, and by adding LLaMA-3-8B-Instruct as an open-source baseline;
> > >
> > > (4) results on LLaMA-3-8B and Qwen3-8B will be incorporated to demonstrate generalizability beyond LLaMA-2 (see **Table R3-A** in the rebuttal for detailed PG-19 perplexity results under standard full attention).
> > >
> > >
> > > > **Regarding limitations**
> > >
> > > Thank you for highlighting this broader impact issue, and also thanks for the insightful comment on broader. We agree that the limitations of stronger and cheaper long-context processing were under-discussed in our submission. Due to the limited space in the rebuttal, our previous response only addressed this point briefly. We expand on it here.
> > >
> > > One limitation is that improving the efficiency of long-context processing may reduce the operational barrier for producing or processing highly coherent long-form content at scale. Although HiCI is not designed for deceptive use, this increased capability could still be misused in adversarial settings. A second limitation is that stronger long-context usability may encourage users to interact with LLMs using increasingly large and sensitive corpora, which can amplify privacy and data-exposure risks in real deployments.
> > >
> > > More broadly, HiCI improves context utilization, but it is not a safety or privacy mechanism. These broader risks therefore require system-level safeguards. We will make this discussion explicit in the final version.
> > >
> > > We sincerely appreciate your thoughtful feedback. Please let us know if there are any remaining concerns we can further clarify. If these clarifications are helpful, we would be grateful if you could consider them in your final evaluation.

---

### Decision · Program_Chairs · 2026-04-30

**Decision:**

Accept (regular)

**Comment:**

This paper proposes HiCI, a hierarchical attention module for long-context language modelling that draws on cognitive theories of discourse comprehension. The module constructs segment-level representations, integrates them into a shared global context, and broadcasts both to condition segment-level attention. The approach is validated through parameter-efficient adaptation of models in the LLaMA-2 family, significantly extending their original context window.

All reviewers recommend acceptance and consistently praised the clarity and motivation of the paper, the soundness of the experimental methodology, and the breadth of the evaluation across language modelling, retrieval, and instruction-following benchmarks. The ablation study was also noted as providing solid empirical support for the key architectural choices.

Several weaknesses were also raised, which I briefly summarise here. First, reviewers noted that the global integration may be non-causal, and called for a more explicit discussion of causal masking and its implications for the reported perplexity results. Second, the paper's positioning within the broader literature could be strengthened, particularly with respect to related work on Global Workspace-inspired architectures and hierarchical or latent-space approaches to long-context modelling. Third, the comparison with closed-source baselines relies on models that are now considerably outdated. Fourth, the paper lacks an efficiency analysis, and does not include a quantitative comparison with training-free hierarchical methods. Finally, the generalisation of HiCI beyond LLaMA-2 remains to be demonstrated. These concerns were partially addressed during the rebuttal with, for instance, results with newer models as well as a compute analysis.

Let me also note that the following reference:

Azerbayev, Z., Tang, J., Huang, Y., Li, Y., Xu, F. F., Wang, Y., Zhou, M., and Deng, J. Proof-pile: Analyzing mathematical reasoning in language models. arXiv preprint arXiv:2204.12672, 2022.

Was flagged as potentially hallucinated by the hallucination detector the conference is using. I'd ask the authors to fix it in future iterations of the manuscript.

Reviewers unanimously share the opinion that strengths outweigh weaknesses. We thus recommend acceptance.